# Transgenerational epigenetic effects imposed by neonicotinoid thiacloprid exposure

Ouzna Dali, Shereen D'Cruz, Louis Legoff, Mariam Diba Lahmidi ⓘ, Celine Heitz, Pierre-Etienne Merret ⓘ, Pierre-Yves Kernanec, Farzad Pakdel, Fatima Smagulova ⓘ

**Neonicotinoids are a widely used class of insecticides that are being applied in agricultural fields. We examined the capacity of a neonicotinoid, thiacloprid (*thia*), to induce transgenerational effects in male mice. Pregnant outbred Swiss female mice were exposed to *thia* at embryonic days E6.5–E15.5 using different doses. Testis sections were used for morphology analysis, ELISAs for testosterone level analysis, RT–qPCR and RNA-seq for gene expression analysis, MEDIP-seq and MEDIP–qPCR techniques for DNA methylation analysis, and Western blot for a protein analysis. The number of meiotic double-strand breaks and the number of incomplete synapsed chromosomes were higher in the *thia* 6-treated group of F3 males. Genome-wide analysis of DNA methylation in spermatozoa revealed that differentially methylated regions were found in all three generations at the promoters of germ cell reprogramming responsive genes and many super-enhancers that are normally active in embryonic stem cells, testis, and brain. DNA methylation changes induced by *thia* exposure during embryonic period are preserved through several generations at important master regulator regions.**

## Introduction

Spermatogenesis is the process by which spermatozoa are produced from germ cells in the testis. In the mouse embryonic testis, germ cells mitotically divide and arrest on embryonic day 14 (Western et al, 2008). Primordial germ cells reach the genital ridge at E10.5, colonizing the gonad at E11.5 (Lovell-Badge, 1993) and undergoing demethylation during E12.5–E13.5 (Hajkova et al, 2002), which is followed by de novo methylation from E13.5 to E16.5. At E16.5, de novo methylation at most of the sites is completed (Lees-Murdock et al, 2003; Aravin et al, 2007; Kuramochi-Miyagawa et al, 2008); however, a group of highly methylated genes demethylated during germline epigenetic reprogramming shows progressive transcriptional activation (Hill et al, 2018). These genes play essential roles in the establishment of the germ cell population. The changes in DNA methylation levels only were not sufficient to induce the expression of these genes. The combined loss of DNA methylation and polycomb repressor complex 1 with further TET1 activation led to germline responsive reprograming (GRR) genes activation, suggesting that activation of GRR genes is a complex process regulated by several mechanisms (Hill et al, 2018). Many GRR genes have meiotic functions (e.g., *Brdt*, *Dazl*, *Ddx4*, *Sycp1*- *Sycp3*, *Rad51c*, and *Hormad1*) (Hill et al, 2018). Consequently, the perturbation of their epigenetic status during development, if persistent, could affect meiosis during males' adult life.

The process of spermatogenesis resumes after birth and starts with the mitotic division of spermatogonia cells. The divided cells further differentiate into a primary spermatocyte. Tetraploid primary spermatocytes divide in a process known as meiosis to give rise to secondary diploid spermatocytes. Then, each secondary spermatocyte undergoes a second meiotic division to give rise to haploid spermatids. Haploid spermatids are further modified to become compact spermatozoa. It is suggested that spermatozoa and oocytes could be implicated in transgenerational inheritance, complex phenomenon, when acquired phenotypic alterations could be transmitted to subsequent generations. One of the first pieces of evidence of parental inheritance in mice came from the experiments with the *agouti* locus. A transcription originating in an intra-cisternal A particle (IAP) retrotransposon inserted upstream of the *agouti* gene causes ectopic expression of agouti protein (Morgan et al, 1999). This epigenetic effect is of maternal origin only and results from incomplete erasure of an epigenetic modification when a silenced *Avy* allele is present in the female germ line. Another study showed that the epigenetic state at *Axin* (Fu) can be inherited transgenerationally via both maternal and paternal germlines (Rakyan et al, 2003). The evidence of DNA methylation preservation has been shown by performing the microinjection of methylated embryonic stem cells to a pregnant mouse, which led to exhibited abnormal metabolic phenotypes in their progeny. Notably, acquired methylation was maintained and transmitted across multiple generations (Takahashi et al, 2023). Besides DNA methylation, histone modifications could be also involved in the mediation of transgenerational effects. The role of sperm histone H3

University Rennes, EHESP, Inserm, Irset (Institut de Recherche en Santé, Environnement et Travail) - UMR_S 1085, Rennes, France

Correspondence: fatima.smagulova@inserm.fr

lysine 4 trimethylation was shown in a genetic mouse model of transgenerational epigenetic inheritance (Lismer et al, 2020).

This study is a continuation of our previous research (Hartman et al, 2021), in which we assessed the effects of different doses of *thia* on the male reproductive system. In a previous study, we showed that exposure to *thia* during gestation leads to spermatogenic defects such as increase in testis–body weight ratios, a decrease in lumen size in seminiferous tubules, synapsing meiotic errors, and increase in the number of double-strand breaks (DSBs). We also observed decreased H3K9me3 levels which were accompanied by increase in retroelement gene expression. Altogether, these effects led to a decrease in spermatozoa numbers (Hartman et al, 2021).

In the present study, we investigated the possibility for neonicotinoid thiacloprid to induce transgenerational effects through ancestral embryonic paternal exposure, focusing particular on a hypothetical role of sperm DNA methylation in those potential effects.

Neonicotinoids have received much attention because of the dramatic decline in bee and bumblebee populations worldwide (Whitehorn et al, 2012; Rundlof et al, 2015). In countries where neonicotinoids are still in use, their presence is very often detected in human samples. For example, a study showed that approximately half of the U.S. general population aged 3 yr and older were exposed to neonicotinoids (Ospina et al, 2019). Notably, young children may experience higher exposure levels than adults (Ospina et al, 2019). Thus, understanding the effects of neonicotinoids is important to a better strategy or therapies for preventing of undesirable effects in exposed humans.

In this study, we show that ancestral thiacloprid exposure induces defects in the reproductive system of third-generation males, perturbs meiosis, causes changes in testis morphology and testosterone levels, and decreases spermatozoa number. We determine that DNA methylation is globally affected in the spermatozoa of three generations of males, with a small proportion of regions differentially methylated in all three generations. We observe that DNA methylation is largely modified at regions overlapping GRR and SE elements. We also report that DNA methylation changes are associated with gene expression changes only at a limited number of loci in directly exposed F1 and non-directly exposed F3 embryonic testes.

# Results

This study is aimed at revealing the transgenerational effects of *thia*. We chose the developmental window from embryonic days 6.5 to E15.5 because of its importance in germ cell program establishment. The mice breeding was described in the Materials and Methods section "Mouse treatment and dissection." The design of the experiments is presented in Fig S1. Notably, breeding was done only through the male line after *thia* exposure, crossing control and thia-derived males to unexposed unrelated females. Three doses of *thia* (0, 0.6, and 6 mg/kg/day) were analysed. Doses were chosen based on the daily authorized dose (0.06 mg/kg/day) of another neonicotinoid, imidacloprid, established by the French Agency for Food, Environmental and Occupational Health and Safety (ANSES)

and were 10 and 100 times higher than the authorized dose. Male progeny was studied at three developmental stages. We analysed embryonic testes at E15.5 to reveal the effects of *thia* on gene expression and DNA methylation during the formation of the population of germ cells. F3 adult progeny mice were analysed for morphological changes at 35 d; at this age, the first wave of spermatogenesis is accomplished. DNA methylation and testosterone levels were analysed at 2 mo; at this age, the reproductive system is fully mature.

## The effects on body and organ weights and testosterone levels

We determined that gestational exposure to *thia* does not significantly affect the body weight in 35-d-old male progeny for dose 0.6 mg/kg/day in the F3 generation (Fig S2A and B). However, mice treated with 6 mg/kg/day tended to have a decrease in of ~5% in the F3 generation (Fig S2A and B). Testis-to-BW ratios did not change in F1 (Fig S2C) but significantly decreased by 10% with 6 mg/kg/day in the F3 generation (Fig S2D), suggesting that in F3, ancestral exposure to *thia* could induce reproductive defects. Thus, importantly, the more profound transgenerational effects are induced only at the high dose of 6 mg/kg/day. We observed a significant decrease in epididymis-to-BW ratios in F1 (Fig S2E) but no significant changes were detected in F3 (Fig S2F). A quantitative analysis of spermatozoa numbers indicated a 29% reduction of spermatozoa in F3 for 6 mg/kg/day (Fig 1A). Nonsignificant changes were detected for 0.6 mg/kg/day in F3 (Fig 1A).

To reveal the effects on germ cell density, we prepared paraffin sections from 35-d-old mice and stained sections with hematoxylin and eosin (Fig S3A). We counted the number of cells in tubules that were in stage VI or VII and measured the total number of cells in tubules per square micrometer (μm2) of the tubule area. The results were averaged and are presented as the number of cells per 1,000 μm$^2$. No significant changes were observed in F3 for 0.6 mg/kg/day. However, for 6 mg/kg/day, a 12% tendency to decrease was observed (Fig S3B). No significant changes in lumen size were observed in F3 for either tested dose (Fig S3C).

Next, we measured the serum testosterone level. We collected blood at the time of the mouse dissection, separated the serum and blood cells by centrifugation, and performed the analysis of testosterone levels in serum by using an ELISA kit. Analysis was performed only for 6 mg/kg/day in F1 and F3 males. The analysis revealed no significant changes in F1 (Fig 1B), but a significant 29% decrease in testosterone in F3 for 6 mg/kg/day was detected (Fig 1C).

In conclusion, gestational exposure to *thia* leads to a decrease in spermatozoa numbers in F1 for both exposure doses, whereas in F3, a spermatozoa decline was observed in the mice treated with 6 mg/kg/day only. This finding suggests that the effects promoted by exposure to 6 mg/kg/day, but not to 0.6 mg/kg/day, were conserved in F3. Therefore, further analyses of the effects of ancestral exposure to *thia* were performed in animals treated with 6 mg/kg/day.

## The transgenerational effects on testis cell populations in F3

To further assess the effects of thiacloprid on germ cell populations, we immunostained paraffin sections with an anti-γH2AX antibody, a marker of double-strand DNA breaks (DSBs), which are

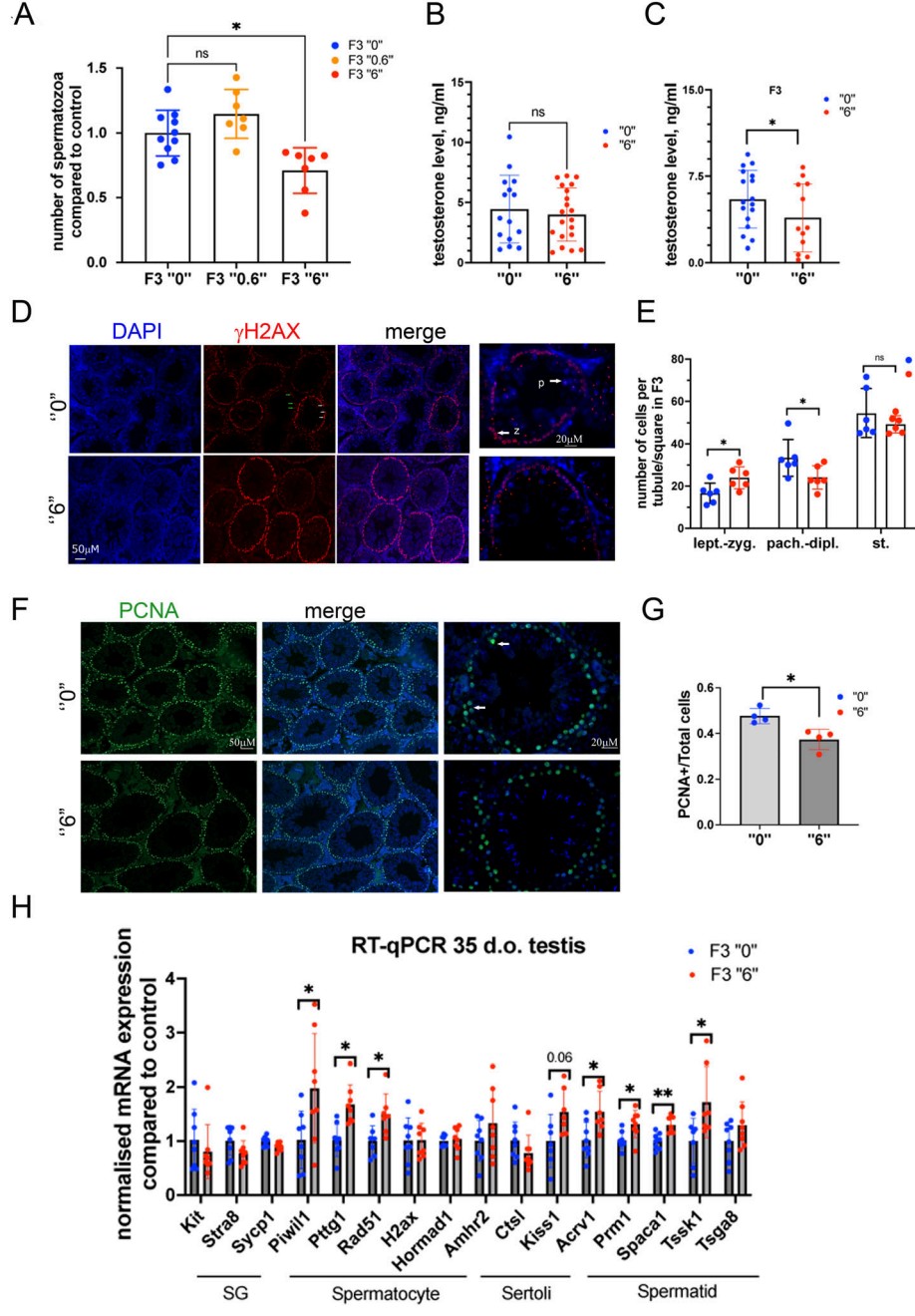

**Figure 1. Effects of ancestral thiacloprid exposure on the reproductive system.**
**(A)** Spermatozoa numbers in F3 males. Spermatozoa were counted in the epididymis of 2-mo-old F3 male mice, the data were averaged and compared with the control, n = 10 dose "0," n = 7 dose "0.6," n = 7 dose "6," *P < 0.05, Kruskal–Wallis test. **(B, C)** Testosterone levels (B) in F1 and (C) in F3 blood serum of 2-mo-old males, F1, n = 20 dose "0," n = 23 dose "6"; F3, n = 20 dose "0," n = 14 dose "6," *P < 0.05, nonparametric Mann–Whitney test. **(D)** A representative image of paraffin 35-d-old F3 male testis section immunostained by γH2AX in control (top) and in the *thia 6*-derived (bottom) mice, bar is 50 μM, images were taken using 20x objectives, the last images are 63x magnification images, z, zygotene cells, p, pachytene cells, bar is 20 μM. **(E)** Quantitative analysis of cells in seminiferous tubules in 35-d-old testis of F3 males, n = 6 dose "0," n = 6 dose "6," *P < 0.05, nonparametric Mann–Whitney test. **(F)** A representative image of paraffin 35-d-old testis section immunostained by PCNA in F3 control (top) and in the F3 *thia 6*-derived (bottom) mice. PCNA protein presents in leptotene–zygotene and pachytene–diplotene spermatocytes in mouse (white arrow), bar is 50 μM. The last images are 63x magnification images, white arrows indicate a PCNA-positive cells, bar is 20 μM. **(G)** Quantitative analysis of PCNA-positive cells in seminiferous tubules in 35-d-old testis F3, n = 4 dose "0," n = 4 dose "6," *P < 0.05, nonparametric Mann–Whitney test. **(H)** Quantitative RT–qPCR analysis of germ cell population markers in 35-d-old testis F3 males, n = 8 dose "0," n = 8 dose "6." *P < 0.05, **P < 0.01, nonparametric Mann–Whitney test. All plots are averaged values ± SD, SG, spermatogonia.

common in meiotic cells. The γH2AX signal was used to identify the individual stages of cells within the germ cell population. Indeed, the γH2AX signal appeared as very bright nuclear staining at the early leptotene–zygotene stage and as a bright spot in sex chromosomes at the zygotene–diplotene stages (Fig 1D). The staining is normally absent in haploid cells such as round spermatids. We performed analysis in the control and in the male progeny in the 6 mg/kg/day group. The spermatids were identified by DAPI staining. In F3, we observed a significant 44% increase in *leptotene–zygotene* cells and a 28% decrease in *pachytene–diplotene* cells (Fig 1E).

In the present study, to evaluate the total number of meiotic cells, we performed the analysis of PCNA marker. PCNA staining was detected previously in spermatogenic cells in the zygotene and pachytene spermatocytes (Chapman & Wolgemuth, 1994). It is suggested that PCNA plays a role in class-switch recombination during meiosis (Roa et al, 2008). We performed staining using paraffin sections (Fig 1F) and analysed the ratio of PCNA-positive cells compared with the total cell number per seminiferous tubule. Our analysis showed a slight but significant decrease in PCNA-positive cells per tubule, which may reflect the decreased number of meiotic cells in exposed progeny in F3 (Fig 1G).

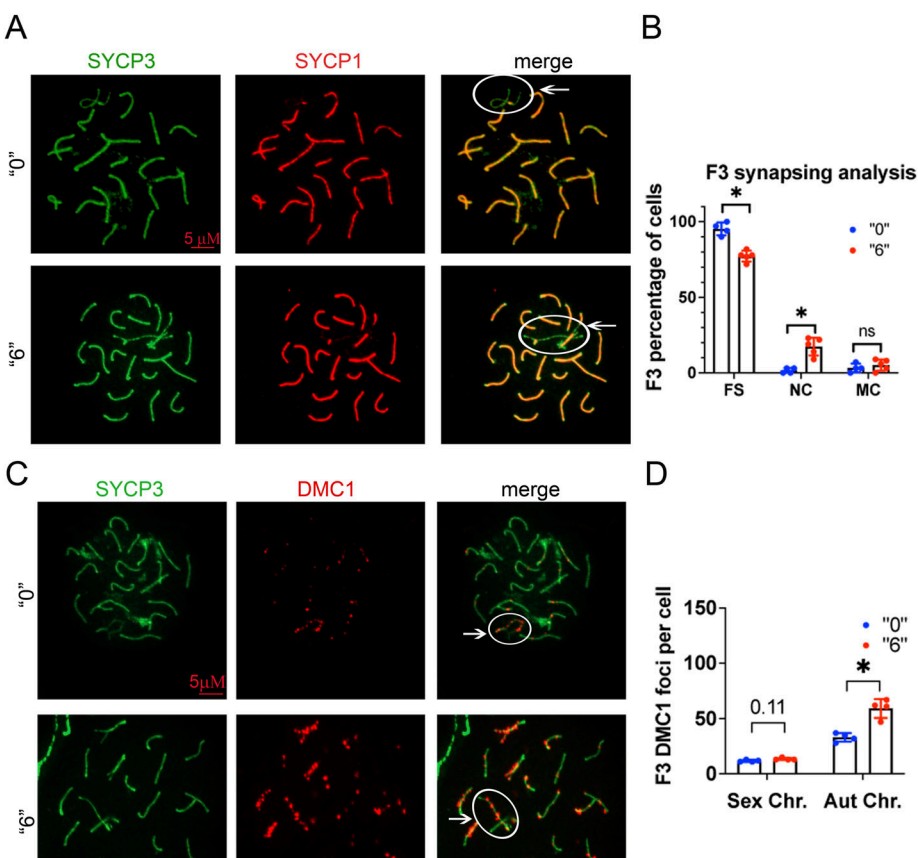

**Figure 2. Meiotic defects in exposed mice.**
**(A)** Representative image of 35-d-old F3 male testis cell spreads immunostained against central, SYCP1 (red) and lateral, SYCP3 (green) components of the synaptonemal complex, white circle indicates sex chromosomes. **(B)** Quantitative analysis of the synapsing defects in 35-d-old testis, FS, fully synapsed, NC, not complete, MC, multiple connections, n = 4 dose "0," n = 4 dose "6," non-parametric Mann-Whitney test, bar is 5 μM. **(C)** Representative image of pachytene stage immunostained by DMC1 (red) and SYCP3 (green) in control (top) and *thia* 6-derived (bottom) 35-d-old testis cell spreads, bar is 5 μM. **(D)** Quantitative analysis of DMC1 foci in F3 35 d-old generation males, sex and autosome (Aut.) chromosome (Chr.) foci were counted independently and compared with control samples, n = 4 dose "0," n = 4 dose "6," *P < 0.05, nonparametric Mann–Whitney test. All plots on the figure represent an averaged value ± SD.

To further evaluate cell type-specific changes, we performed RT–qPCR analysis using cell type-specific markers. We performed analysis for markers specific for spermatogonia (SG), spermatocytes (SC), spermatids (ST) and Sertoli cells. The analysis was done in whole testis. As a marker database source, we used single-cell sequencing data (Green et al, 2018). In F3 thia-derived mice, we observed that spermatocyte-specific genes such as *Piwil1*, *Ptgg1*, and *Rad51* had increased expression (1.9, 1.7, and 1.5 times, respectively) compared with controls. Genes specific for the spermatid fraction, *Prm1*, *Spaca1*, and *Tssk1*, had increased expression (1.3, 1.3, and 1.7 times, respectively). These data suggest a possible impact of *thia* exposure on gene expression in both meiotic and postmeiotic populations of cells (Fig 1H).

Thus, gestational exposure to thiacloprid leads to changes in testis morphology, affects testosterone levels, alters testis-specific gene expression in adults, and leads to spermatozoa decline, suggesting that ancestral *thia* exposure has a deleterious effect on reproductive function in third-generation males.

### Synapsing defects and persistence in meiotic DSBs and chromosomes were observed after ancestral exposure to *thia*

Because we observed synapsing defects in F1 generation males (Hartman et al, 2021), we investigated the synapsing efficiency between the control and treatment groups in F3. We immuno-stained the surface testis spread slides against SYCP3 (meiotic chromosome marker) and SYCP1 (a marker of completely synapsed chromosomes) (Fig 2A). SYCP1 and SYCP3 were analysed in a minimum of 35 *pachytene* cells from the control and exposed groups. In normal meiosis, SYCP1 and SYCP3 completely overlap except for the sex chromosomes, and in pathological conditions, SYCP1 and SYCP3 have regions that do not completely overlap. In F3, we observed an 11fold increase in incomplete synapsed chromosomes (Fig 2B).

To assess DNA repair efficiency between the control and treatment groups, we immunostained the slides against SYCP3 and DMC1 (a marker of meiosis-specific recombinase that binds to DSBs) proteins (Fig 2C). DMC1 foci were enumerated for the control and exposed groups. We chose cells that were in the early *pachytene* stage (defined as *pachytene* based on Page et al [2012]), visually fully synapsed, and showed more than 10 DMC1 foci. We counted the DMC1 foci in sex and autosomal chromosomes separately to determine whether there was a chromosome-specific alteration. In F3, we observed a 1.8fold increase in the number of DMC1 foci in autosomes and a 1.1fold tendency to increase for sex chromosomes (Fig 2D).

In conclusion, we observed meiotic defects in F3 that were previously found in F1, namely, the persistence of DSBs and a higher number of synapsing errors, suggesting that meiosis was

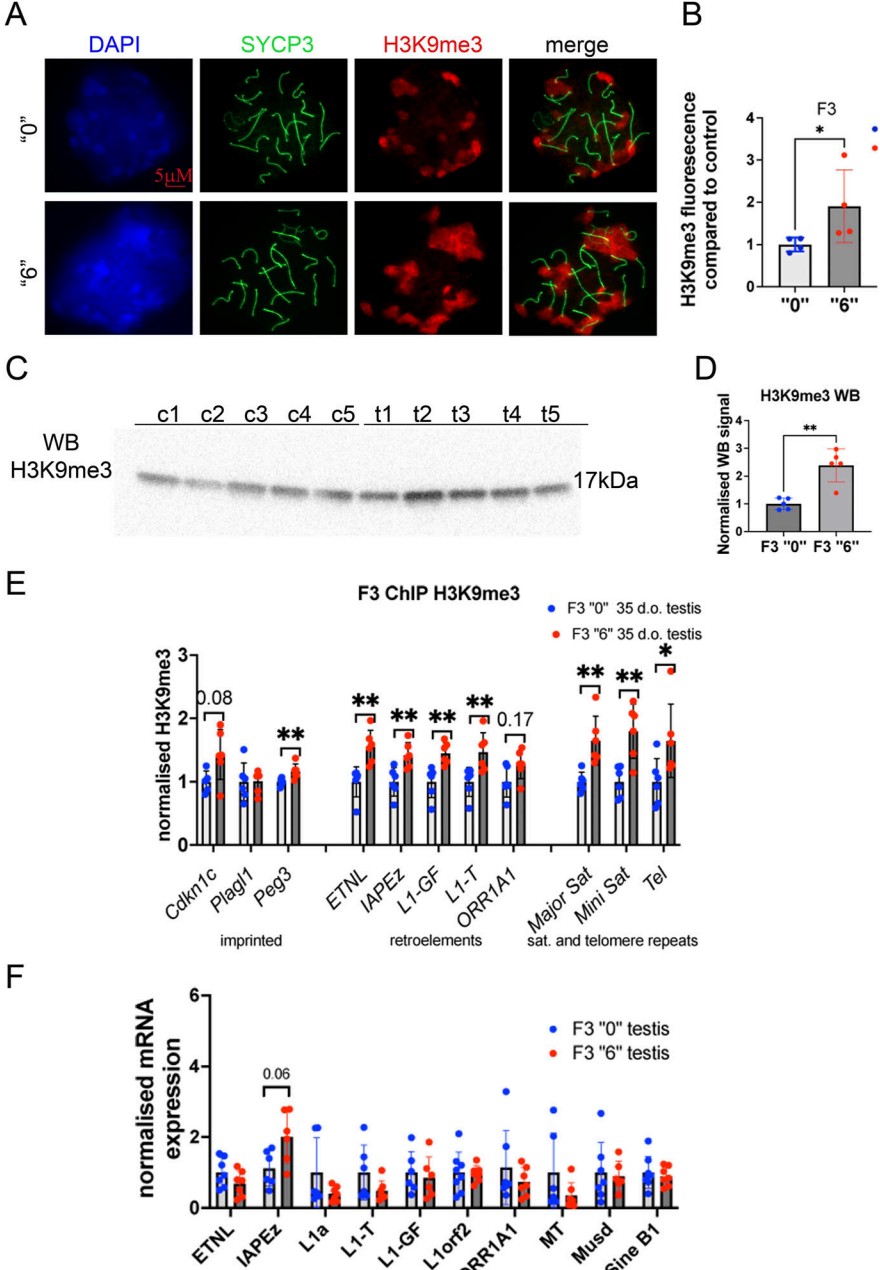

**Figure 3. The effects of ancestral thiacloprid exposure on H3K9me3 levels and retroelement transcription activity.**
**(A)** Representative image of 35-d-old F3 male testis spreads immunostained against H3K9me3 (red) or SYCP3 (green), bar is 5 μM. **(B)** Quantitative analysis of H3K9me3 intensity in F3 generation males, n = 5 dose "0," n = 4 dose "6," *P < 0.05 nonparametric Mann–Whitney test. **(C)** Representative image of H3K9me3 Western blot of testis tissue, c1-c5 are control, t1-t5 are *thia* -derived samples. **(D)** Quantitative analysis of Western blot, n = 5 dose "0," n = 5 dose "6," **P < 0.01 nonparametric Mann–Whitney test. **(E)** Quantitative ChIP–qPCR analysis of H3K9me3 in 35-d-old testis of F3 males, n = 5 dose "0," n = 5 dose "6," *P < 0.05, **P < 0.01 nonparametric Mann–Whitney test. **(F)** The analysis of retroelement expression in 35-d-old testis of F3 males by RT–qPCR, n = 7 dose "0," n = 7 dose "6." All plots on the figure represent an averaged values ± SD.

perturbed in both F1 and F3 generations after in utero *thia* exposure.

## The effects of *thia* exposure on H3K9me3

Because we previously observed decrease in H3K9me3 (Hartman et al, 2021), an important mark for heterochromatin organization, in F1, we decided to evaluate the effects of *thia* exposure on H3K9me3 levels in F3. We immunostained spreads against H3K9me3 and SYCP3 and performed quantitative analysis of H3K9me3 in individual cells (Fig 3A). In F3, the normalized values for H3K9me3

were almost two times higher in the descendants of the exposed mice than in those of the control mice (Fig 3B).

To analyse the global level of H3K9me3, we extracted the histone proteins of 35-d-old male testis, loaded the proteins on a 4–15% gradient SDS–PAGE gel, and blotted against H3K9me3 (Figs 3C and S4A). The band intensity was normalized to the Ponceau stain shown in Fig S4B. The quantitative analysis of WB showed a significant increase in H3K9me3 level in F3 testis (Fig 3D), which is consistent with immunofluorescence data.

To reveal the level H3K9me3 occupancy at the regions which are normally enriched in H3K9me3, for example, imprinted genes, retroelements, satellite and telomere repeats, we performed ChIP

followed by qPCR. We determined that H3K9me3 has an increased occupancy at imprinting gene *Peg3*, at retroelements, at the minor and major satellites and at telomere repeats (Fig 3E). This result is consistent with immunofluorescence and WB data.

We analysed the level of retroelement expression by RT–qPCR in the testis of 35-d-old F3 males. Expression of retroelements in F3 did not change by ancestral exposure to *thia*, with only a tendency for IAPez to increase in expression (Fig 3F).

Our data show that the global level of H3K9me3 is increased in F3 adult 35-d-old testis upon ancestral *thia* exposure and this change in H3K9me3 does not significantly affect the expression of retroelements.

### RNA-seq analysis in embryonic thia-derived animals reveals effects on protein-coding genes important for meiosis, cell adhesion, and central nervous system development

To explore the molecular mechanisms of *thia* action, we performed transcriptomic analysis of mRNA using paired-end stranded RNA sequencing using three biological replicates for the *thia*-derived and control embryonic E15.5 testes. Each replicate was a pool of testes from three embryos because of the small size of this organ. Principal component analysis and dispersion heatmap plots show substantial variation between F1 replicates (Fig S5A and B), which is probably the reason why only a small number of differentially expressed genes (DEGs) were detected in F1. We identified 198 DEGs in F1 (FDR < 0.15) (Figs 4A and S5C). Among the DEGs, we identified 101 genes that were down-regulated and 97 that were up-regulated. We performed a functional annotation of the down- and up-regulated genes separately by using the Gene Ontology program DAVID (Fig 4B), with genes which are expressed in embryonic testis as a background list. In F1, among the down-regulated genes, we observed enrichment in germ cell development, male meiosis, and transcription repressor-related genes. Among up-regulated genes, the strongest enrichment was found in the genes related to sensory perception of the sound, muscle contraction, cell differentiation, and skeletal system development (Fig 4B). In F3, we identified less variations between replicates compared with F1 (Fig S5D and E). We determined that 2,277 genes were differentially expressed in F3 (FDR < 0.1) (Figs 4C and S5F). In F3, among DEGs, we identified 882 genes that were down-regulated and 1,395 that were up-regulated. The down-regulated genes were related to translation, rRNA processing, protein folding, and mitochondrial electron transport. The up-regulated genes were the genes that encode proteins for cell adhesion, cell migration, angiogenesis, axon guidance, and cholesterol metabolic process-related functions (Fig 4D).

We confirmed by RT–qPCR the expression of genes which were found to be differential in F3 RNA-seq data. The genes of interest are known to be implicated in steroid hormone signalling. Out of eight tested genes, five of them showed changes similar to those found in the RNA-seq experiment (Fig 4E). RNA-seq and RT–qPCR methods do not use the same algorithm for the analysis. RNA-seq is based on the counting of sequencing reads over the region; whereas RT–qPCR method is amplifying the region restricted by the primers. Both methods however show similar directions in changes in our experiments. We observed a significant increase in *Inha*

expression, a decrease in *Dkk1* and tendencies toward an increase in *Cyp17a1*, *Scarb1*, and *Star* (Fig 4E).

In conclusion, gestational exposure leads to a global change in the expression of many developmental genes relevant to nervous and reproductive system development. Fewer genes were significantly affected in F1 than in F3 because of higher dispersion between biological replicates in F1 than in F3.

### Sperm DNA methylation analysis showed that gestational *thia* exposure leads to global alteration in directly exposed F1 and F2 and in indirectly exposed F3 male spermatozoa

Because spermatozoa and oocytes are the only cells that are transmitted to the next generations, we investigated whether DNA methylation was affected in the spermatozoa of mice. To this end, we performed DNA methylation analysis using a genome-wide sequencing technique. First, we isolated the mobile fraction of spermatozoa from the epididymis: immediately after dissection, the epididymis was pierced in the cauda, placed in DMEM without adding factors or serum, and incubated at 37°C for 1 h to allow spermatozoa to swim out. We verified the quality of spermatozoa by light microscopy. The released spermatozoa were filtered, transferred to a new tube, pelleted by centrifugation, and kept at –80°C until use. Spermatozoan DNA was extracted using a modified protocol described in the Materials and Methods section. For each MEDIP assay, we used spermatozoan DNA from a single mouse. We performed MEDIP using a minimum of three biological replicates for the control and treatment groups of each generation. Methylated DNA was enriched by using the EpiMark Methylated DNA Enrichment Kit as described in the Materials and Methods section. The DNA concentration was measured and equal amounts of MEDIP and input were taken for sequencing library construction. The generated libraries were subjected to massive parallel sequencing on the Genomic platform at Strasbourg. The sequencing data were analysed as described in the Materials and Methods section and differentially methylated regions (DMRs) were identified for each generation. The analysis showed that nearly 120,000 CpG-rich regions were detected and most methylated regions overlapped the genes (Figs S6 and S7A–L).

We identified 1637, 2742, and 1667 DMRs in F1, F2, and F3, respectively (FDR<0.1 for F1 and F2 and FDR<0.12 for F3) (Table 1). Samples showed some variations in all three generations (Fig S8A–C). Among the differential peaks, most of the DMRs in F1 and F2 showed decreased DNA methylation (Fig S8D and E). An increase in DNA methylation in many regions was observed in F3 compared with F1 and F2 (Fig S8F). DMRs are mostly located at promoters, exons, and introns (Fig S9). However, nearly ~30% of DMRs are in distal intergenic regions (Fig S9). Notably, some alterations were present in only one generation. For example, at the gene body of *Dnmt3a* DNA, methylation significantly increased only in F1, in the vicinity of *Sox9* only in F2 and, at the *Gsc* locus, the changes were significant in F3 males only (Fig 5A). To reveal the genes that are differentially methylated in all three generations, we assigned DMRs to genes using GREAT. We found that 1,567, 3,792, and 2,467 genes were differentially methylated in F1, F2, and F3, respectively. We found that 481 genes (9.3%) had DNA methylation changes in all three generations (Fig 5B and Table

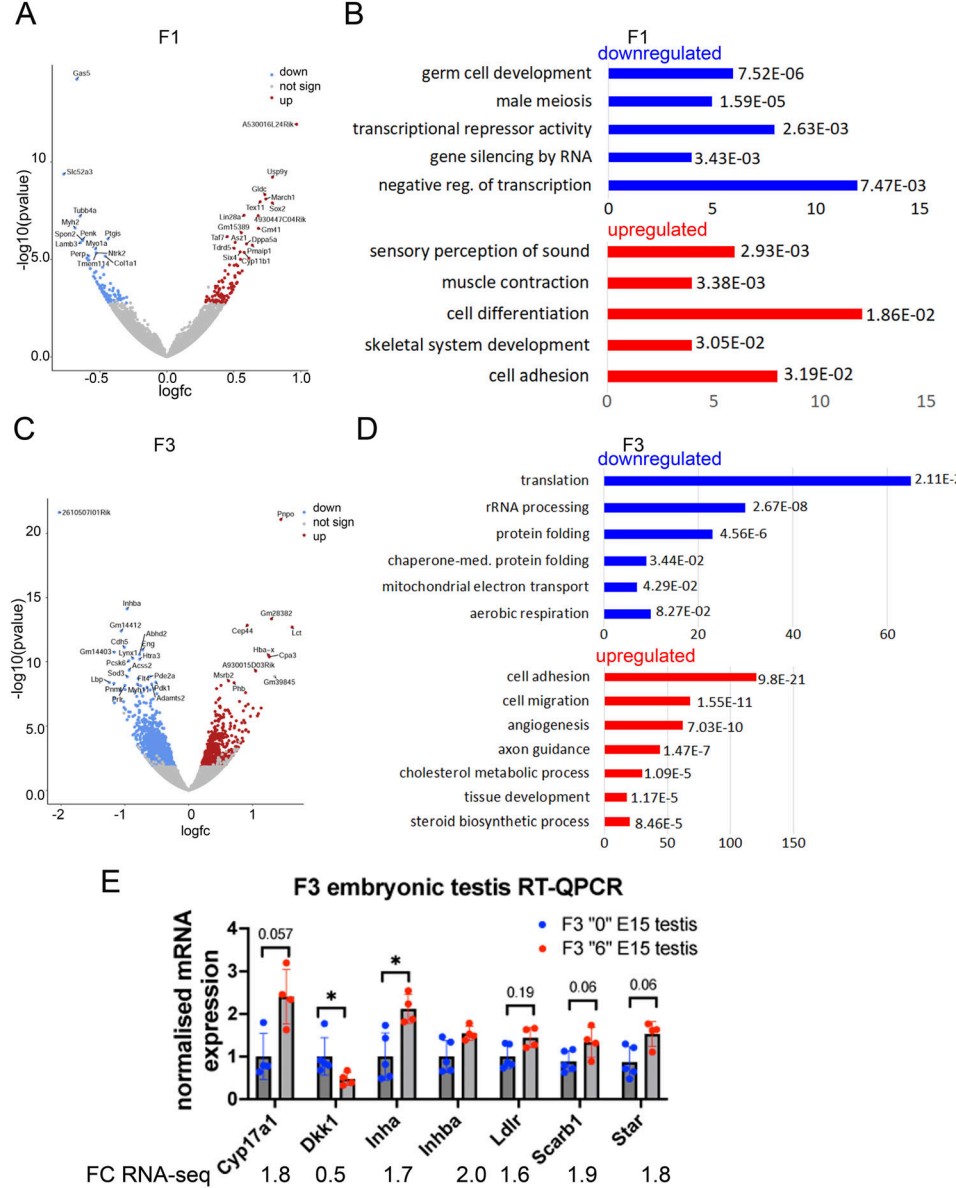

**Figure 4. RNA-seq analysis in embryonic E15.5 F1 and F3 testes.**
**(A)** Volcano plot of differentially expressed genes in F1 testis, down-regulated genes are in blue and up-regulated are in red, n = 3 dose "0," n = 3 dose "6." **(B)** Functional annotation of the down-regulated (blue) or up-regulated (red) differentially expressed genes in F1 embryonic testis were done using DAVID, Fisher's exact test was adopted in DAVID to measure the gene-enrichment in annotation terms, the genes were sorted by *P*-value in F1. **(C)** Volcano plot of differentially expressed genes in F3, n = 3 for dose "0" and dose "6." **(D)** The down-regulated (blue) and up-regulated genes (red) were annotated separately by using DAVID. The genes were sorted by adjusted *P*-value in F3. The bars represent the number of genes in each Gene Ontology group. **(E)** Quantitative RT–qPCR analysis of DEGs in embryonic F3 testis, the RNA-seq fold change (FC) values are indicated under the graph. RT–qPCR plots represents an averaged values ± SD, n = 5 dose "0," n = 4 dose "6." *P* < 0.05, nonparametric Mann–Whitney test.

S1). Functional annotation of these common genes by DAVID showed that they are highly enriched in cell adhesion-, multicellular organism development-, and cell differentiation-related genes (Fig 5C). We performed functional annotation of DMRs for each generation separately with GREAT and found that DNA methylation in F1 males mostly changed in the vicinity of the genes related to the regulation of catenin, rhombomere, and notochord development (Fig 5D). In F2, DNA methylation changed in the vicinity of genes related to synaptonemal complex assembly, somatic stem cell division, and neuroblast division (Fig 5E). In F3, DNA methylation changed in the vicinity of genes related to meiosis (Fig 5F).

We looked closer at the regions with altered DNA methylation in F1 and F3. Despite the fact that most of the DMRs did not have the same direction in changes, we identified 30 regions doing so. GREAT assigned these regions to 36 genes. This group includes developmental genes (*Isl1, Lama1, Tle1, Sox2, Foxn1, Foxd4, Inha, Zfp64*) (Table S2). We suggest that these regions could resist reprograming and preserve their DNA methylation. In addition, most of the 30 genes kept a similar level of fold change in F1 and F3 (Table S2). We decided to verify whether those regions were located near IAP retroelements which could escape DNA demethylation during germ cell reprograming. We found the presence of IAP element *IAP1-MM_LTR* near *Isl1* (less than 300 bp from DMR region). Further experiments are required to confirm the role of the IAP in *Isl1* DNA methylation conservation.

Next, we investigated the state of DNA methylation in a few regions using MEDIP–qPCR and sperm DNA. We performed this analysis using DNA from the sperm of F1 and F3. In F1, most of the targets showed a decrease in DNA methylation similar to what has been determined by MEDIP-seq (*Brdt, Ddx4, Tdrd1, Spo11, Dmrtb1, Nanog, Smad2, Sox2*)

**Table 1. Differentially methylated regions in spermatozoa of the F1–F3 generation.**

| Generation | Up | Down | Total |
|---|---|---|---|
| F1 | 456 | 1,181 | 1,637 |
| F2 | 427 | 2,315 | 2,742 |
| F3 | 1,364 | 303 | 1,667 |

(Fig 5G). We also analysed the DNA methylation state at retroelements and found that there is a tendency for DNA methylation to decrease in F1 sperm (Fig 5G, right plot). In F3, we observed an increase in DNA methylation at most targets (Hormad1, Spo11, Mesdc1, Peg3, Smad, Sox2, Dlc1) (Fig 5H). In contrast to F1, retroelements in F3 showed an increase in DNA methylation (Fig 5H, right plot).

Next, we analysed the DNA methylation state of 45 GRRs. We found that DNA methylation changed at these genes in all three generations of males. Out of 45 known GRRs, we observed that DNA methylation changed in 18, 35, and 26 genes in the F1, F2, and F3 generations, respectively (Table S3), suggesting that most of the GRR regions had perturbed DNA methylation. For example, we detected changes in DNA methylation in the promoters of Ddx4 and Sycp2 (Fig S10).

Superenhancer (SE) elements are other regions that could be important for germ cell population development, as they control the gene expression essential for cell identity establishment (Whyte et al, 2013). SEs were first identified in pluripotent embryonic stem cells (Whyte et al, 2013). A database of SEs was created (Jiang et al, 2019) and recently updated (Wang et al, 2023). It suggested that DNA methylation is involved in the regulation of SE activity. Variable levels of DNA methylation have been found at superenhancers (SEs), and in experiments using allele-specific reporters, it has been reported that the SE DNA methylation state could be dynamically switched (Song et al, 2019). To gain insight into the possible impact of neonicotinoids on these regions, we extracted superenhancer coordinates from the https://www.asntech.org/dbsuper/ database for E15.5 brain, adult testis, and ES cells. We chose to analyse SE that are active in germ cells (testis) and somatic cells (brain) and in precursors of both tissues (ES). First, we identified superenhancers that overlap DNA methylation regions in control samples (Table 2).

We identified that 199 (~50%) out of 400 brain SEs, 123 (~59%) out of 210 testis SEs and 84 (37%) out of 231 ES cell SEs had DNA methylation marks in unexposed sperm, $P < 10 \times 10^{-6}$, hypergeometric test (Tables S4, S5, and S6). Next, we analysed whether some of them are differentially methylated in the spermatozoa of exposed males. We found that SE element methylation was affected in all three generations. For example, the DNA methylation at superenhancers, mSE_00104 located near Mesdc1/Tlnrd1 and mSE_00085 found near Nanog, were altered in all three generations (Fig 6A and Tables S7, S8, and S9).

Next, we asked whether some of the altered sperm DNA methylation regions could correlate with DNA methylation variations in embryonic testes. We performed MEDIP–qPCR analysis of the regions found to be differentially methylated in sperm. We designed primers that amplify the DMR of some GRR genes (Brdt, Ddx4, Hormad1, Tdrd1), genes essential for spermatogenesis (Spo11, Smad2, Syce3, Wnt5a) and superenhancers found in ECS cells, brain, and testis. We performed MEDIP–qPCR using DNA which had been extracted from F1 and F3 pools of several embryonic testis. DNA methylation in F1

embryos decreases in numerous genes, notably in genes associated with spermatogenesis (Brdt, Hormad1, Smad2, Wnt5a) and located near SE (Lif, Fat1, Nanog, Sox2, Dlc1 and Ntn5) (Fig 6B). In contrast to F1, we observed that many spermatozoa DMRs found in F1 showed an increase in DNA methylation in F3 embryonic testes, including Brdt, Ddx4, Dmrtb1, Spo11, Tdrd1, Wnt5a, and Hoxa3 (Fig 6C).

Next, we asked to what extent DNA methylation changes contribute to gene expression alterations. To do so, we compared the list of DEGs in embryonic testes with the genes assigned by GREAT in DMRs. In F1, out of 198 DEGs, only 39 (25%) were differentially methylated (Table S10), and in F3, out of 2,277, only 300 (13%) (Table S11) were differentially methylated, suggesting that embryonic testis gene expression changes and adult sperm DNA methylation changes were poorly correlated. The DEGs in embryos showed a correlation with lipid binding (Nr5a2, Ogt), somatic stem cell population maintenance (Nanog, Sox2, Sall4), and negative regulation of transcription (Sox2, Sall1, Sall4, Zfp131, Nanog) in F1 males (Fig 6D). In F3, cell adhesion genes (Nectin1, Celsr2, Itga2,8, 9), nervous system development genes (Ngfr, Zmynd8, Efgfr, Zeb1, Disc1), and multicellular organism development genes (Ephb4, Bmp2, Asz1) were among the most enriched (Fig 6E).

We also asked whether the changes in DNA methylation in sperm could be detected in adult testes. To this end, we analysed gene expression in adult 35-day-old male testes by RT–qPCR. We observed that several GRR genes (Brdt, Dazl, Ddx4) had decreased expression in the testes of F1 generation and these genes also had decreased DNA methylation in the sperm of the F1 generation (Fig S11A). In F3, Brdt, Dazl, Ddx4, and Tdrd1 also had increased expression, but DNA methylation was not significantly affected for Brdt and Dazl but increased for Ddx4 and Tdrd1 (Fig S11B), suggesting that genes important for meiosis were deregulated in F3 testes. Similarly, we assessed gene expression in the adult 2-mo-old brain, we analysed the gene expression in cerebellum. We analysed the expression of the genes that have alterations in sperm DNA methylation and play roles in the brain. We found that the expression of Dlc1, Mesdc1/Tlnrd1, Nanog, and Sox2, which are located in the vicinity of SEs, was increased in the F1 brain (Fig S12A), but changes were not significant in the F3 brain, except for Nanog, suggesting that in the adult F3 brain, gene expression was not strongly affected compared with that in the F1 brain (Fig S12B).

Thus, our data show that changes in DNA methylation in the spermatozoa of adult males may not be fully correlated with gene expression in the embryonic testis. A change in ~30 regions occurring at the early embryonic stage was preserved and transmitted to F3 generation.

# Discussion

## Transgenerational effects on spermatogenesis

In this study, we aimed to reveal the effects of a widely used insecticide, thiacloprid, on the male reproductive system and the descendants of exposed individuals. In F3, we found that reproductive parameters such as spermatozoa numbers, density of cells, and testis weight were significantly altered only at the highest dose,

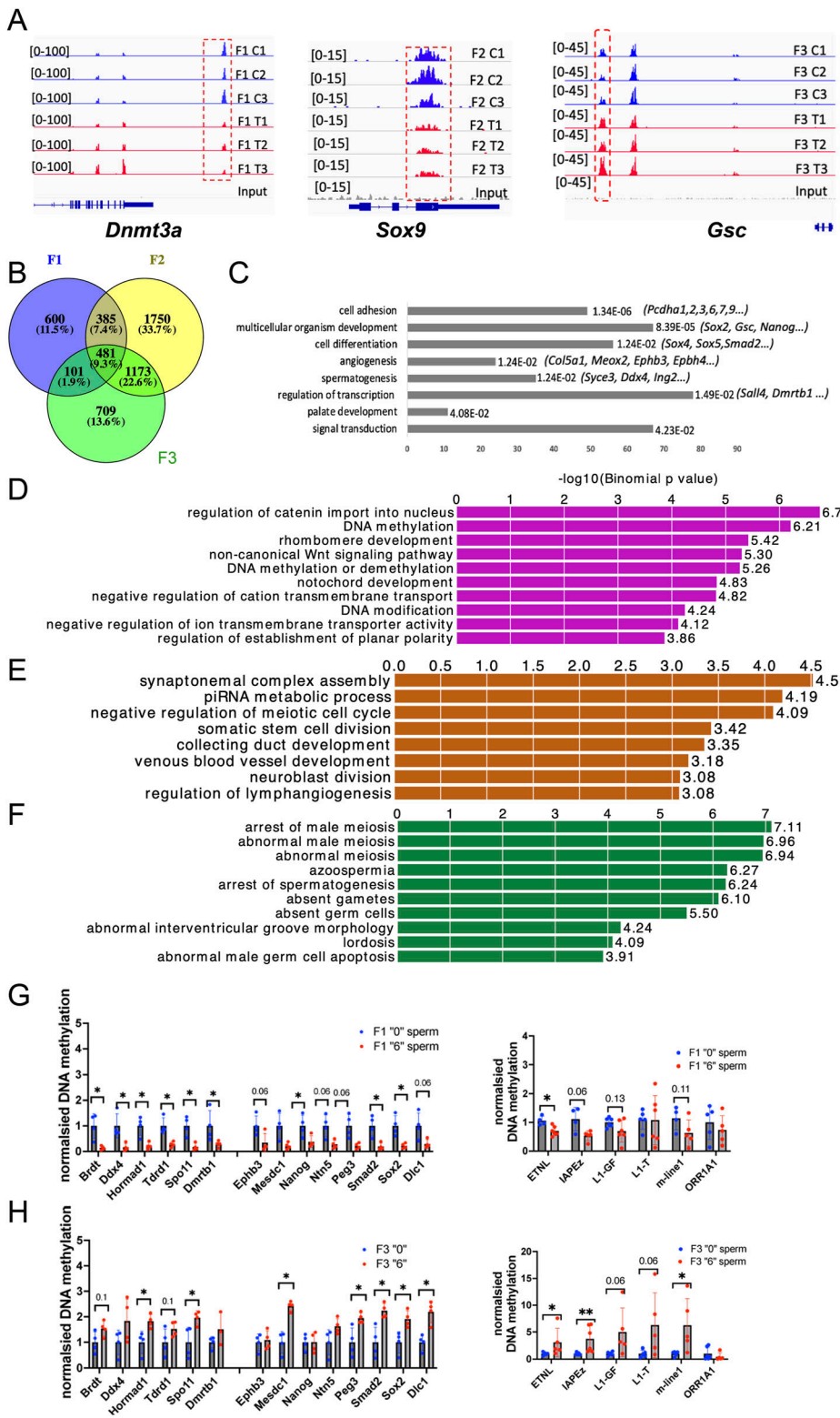

**Figure 5.  Genome-wide DNA methylation analysis in spermatozoa of the 2-mo-old F1, F2, and F3 mice.**
**(A)** A representative image of mapped reads. The differentially methylated peaks near *Dnmt3a*, *Sox9*, and *Gsc* are shown in red dashed boxes. Plots from control samples are shown in blue, plots from *thia*-derived samples are shown in red. Each control and treatment groups contained minimum three replicates. The sequencing reads were mapped to the reference *mm10* genome, normalized, and converted to *bedgraph* files which were visualized in IGV, the signal intensity is shown in brackets, and the differential peaks are marked by dashed box. **(B)** Venn plot represents common genes located in DMRs in F1, F2, and F3. **(C)** Functional annotation of genes located in common DMRs, bars sorted by adjusted *P*-values, and each bar represents the number of genes in each group. **(D, E, F)** Functional annotation "biological process" of genes located in DMRs (D) in F1, (E) in (F2) and (F) in F3 was performed by GREAT. Each bar represents −log 10 (binominal *P*-value) calculated by GREAT. **(G, H)** MEDIP-qPCR analysis in F1 and (H) in F3 E15 embryonic testis. Right plots are MEDIP–qPCR analysis of spermatogenesis and SE genes from spermatozoa of 2-mo-old males. F1, n = 4 dose "0," n = 4 dose "6," F3, n = 4 dose "0," n = 4 dose "6." The right plots represent the analysis of MEDIP-qPCR analysis of retroelements, F1: n = 6, dose "0," n = 6 dose "6"; F3: n = 4 dose "0" and n = 4 dose "6." MEDIP-qPCR plots are averaged MEDIP values ± SD, *P < 0.05, **P < 0.01 nonparametric Mann–Whitney test.

**Table 2.   Superenhancer regions found have DNA methylation in the sperm.**

| Tissue | Total SE | SEs have DNA methylation in sperm | DMRs in F1 | DMRs in F2 | DMR in F3 |
|--------|----------|-----------------------------------|------------|------------|-----------|
| Brain | 400 | 199 (49.8%) | 14 (7%) | 34 (17%) | 12 (6%) |
| Testis | 210 | 123 (58.6%) | 3 (2.4%) | 19 (15.4) | 8 (6.5%) |
| ES cells | 231 | 84 (36.3%) | 3 (3.6%) | 8 (9.5%) | 4 (4.8%) |

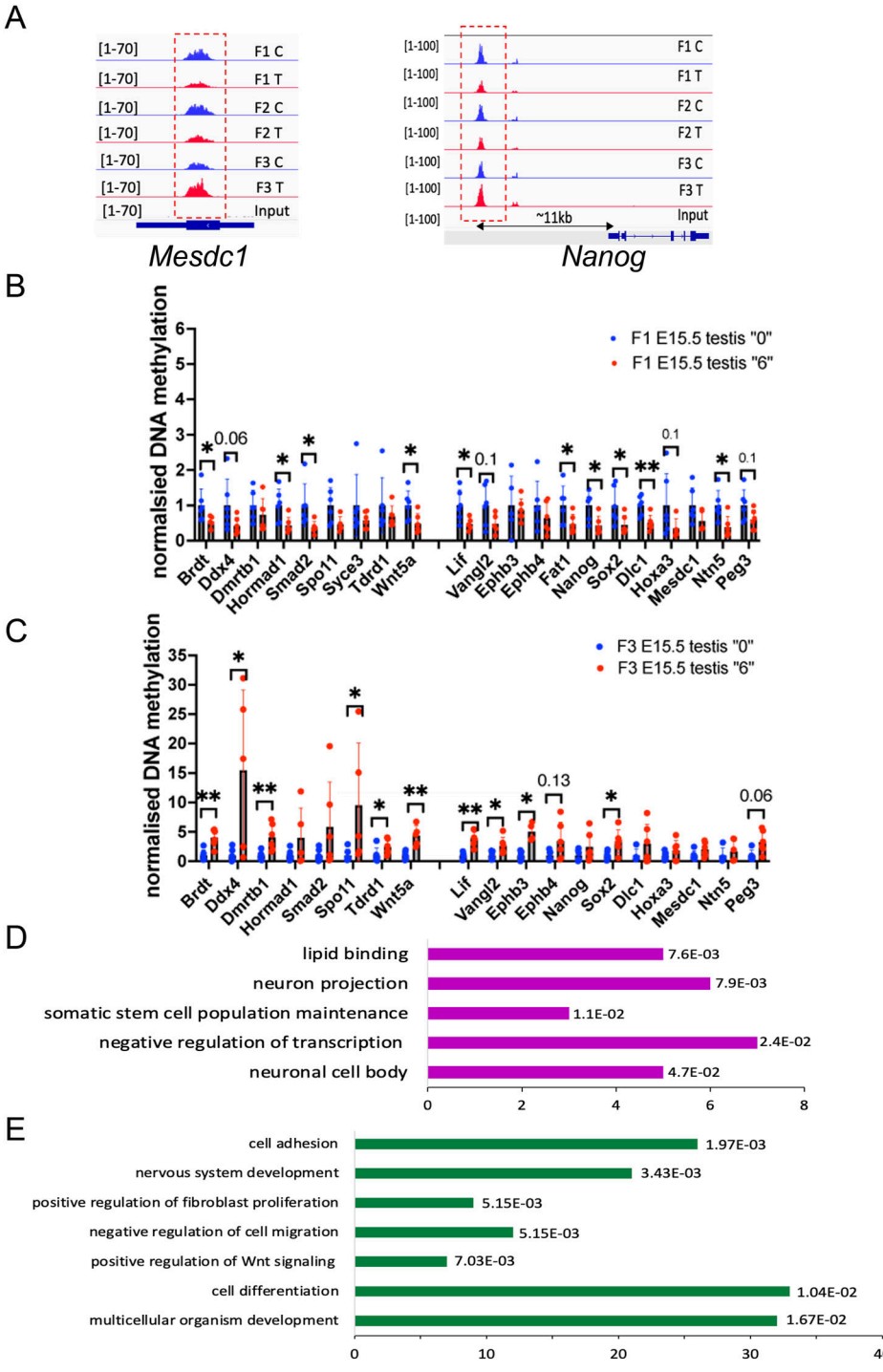

**Figure 6.   DNA methylation of SE and GRR regions changed in spermatozoa.**
**(A)** At the vicinity of regions including superenhancers active in ES cells (*Mescd1, Nanog*). Each plot represents the normalized sequencing reads, the signal was calculated for each sample, and the statistically significant difference was calculated by the Limma test. Overlaid biological replicate plots from the control group are shown in blue, and overlaid biological replicate plots from the treatment group are shown in red. Signal intensity is shown in the brackets. **(B, C)** MEDIP–qPCR analysis in embryonic E15.5 testis (B) in F1 and (C) in F3, F1 and F3 n = 6 dose "0," n = 6 dose "6." MEDIP–qPCR plots represent averaged values of qPCR ± SD, *$P < 0.05$, **$P < 0.01$, nonparametric Mann–Whitney test. **(D, E)** Differentially expressed genes in RNA-seq have DMRs in MEDIP-seq data (D) in F1 and (E) in F3. The genes were sorted by $P$-value (in F1) or by adjusted $P$-value (in F3).

6 mg/kg/day, but not at 0.6 mg/kg/day. We suggest that the effects promoted by 0.6 mg/kg/day in F1 may have reverted in F3. We observed that F3 males have meiotic defects. We believe that the synapsing meiotic defects and persistence of DSBs could lead to a reduction in gamete production. As it was previously shown, p53 and TAp63 participate in the recombination-dependent pachytene arrest and apoptosis in mouse spermatocytes (Marcet-Ortega et al, 2017). The persistence of DMC1 foci were observed in mice model knockout experiments with ablation of meiotic specific proteins such as *Meiob* (Souquet et al, 2013) and *Tex19.1* (Crichton et al, 2018). In these cases, meiosis was arrested at the zygotene/pachytene stage of prophase I and cells failed to progress further and were eliminated. We can similarly suggest that there is a meiotic failure in *thia* progeny of F3 mice that could lead to meiotic arrest and apoptosis that explain the decrease in spermatozoa numbers.

We also observed alterations in H3K9me3 decrease in F1 and increase in F3 testis. H3K9me3 are enriched at centromeres, which are essential for faithful chromosome segregation during mitosis and meiosis. Centromeres organize the kinetochore, the protein machinery that attaches sister chromatids or homologous chromosomes to spindle microtubules and regulates their disjunction (Talbert & Henikoff, 2022). Long-term depletion of centromeric H3K9me3 causes mitotic mis-segregation (Martins et al, 2020). Thus, we can imagine that alterations in H3K9me3 could affect the meiotic division and thereby contribute to decreased meiotic progression.

The changes in meiosis could be because of alterations in the DNA methylation of GRRs, which are activated during embryonic development. Because most of the GRRs are meiotic genes, changes in these genes may perturb meiotic progression. We also observed that testosterone levels had decreased and that the genes essential for testosterone production had higher methylation levels in F3. Testosterone is important for gametogenesis because it is a natural ligand for the androgen receptor. Androgen receptor-deficient mice have failed to produce gametes (Zhang et al, 2006). Thus, changes in DNA methylation could affect gene expression and thereby perturb reproductive function.

### Epigenetic effects promoted by *thia* and possible mechanisms explaining their preservation in subsequent generations

In this study, we aimed to reveal the role of DNA methylation in the transmission of *thia* effects to the next generations. Analysis of sperm DNA methylation showed a global impact on sperm DNA methylation imposed during gestational exposure. The changes in DNA methylation could result from interference with DNA methyltransferase activities during gestational exposure. De novo methylation is established by the DNA methyltransferases DNMT3A and DNMT3B. These enzymes exhibit nonoverlapping functions during development (Okano et al, 1999). Because several reprogramming events occur between the first and the third generations, it is not clear how the induced changes in sperm could be persistent and transmitted to subsequent generations. In mammals, genes that are highly methylated in sperm are rapidly demethylated in the zygote only hours after fertilization, before the onset of the first round of DNA replication (Oswald et al, 2000). As evoked in the introduction, the localization of regions near retroelements could contribute to the resistance to DNA demethylation. We

identified that the DMR region near *Isl1* is localized in close proximity to IAP. Further work is required to establish the role of IAP or other retroelements in the conservation of DNA methylation.

We suggest also that some regions could preserve a "memory" of changes at a certain place via consequent alterations in histone H3K4me3. Indeed, a study on human zygotes showed that active genes with histone H3 trimethylated at lysine 4 (H3K4me3) at promoter regions are essentially devoid of DNA methylation (Guo et al, 2014), suggesting that H3K4me3 prevents DNA methylation in early zygotes. In addition, early embryos tend to retain higher residual methylation at evolutionarily younger and more active transposable elements (Guo et al, 2014), suggesting that some genes could exploit promoters of active transposable elements for their activation. The analysis of DNA methylation in zygotes showed that oocyte contribution to DMRs primarily resides in CpG island-containing promoters, whereas sperm contribution to DMRs was predominantly intergenic (Smith et al, 2012). Thus, it is conceivable that a newly established DNA methylation pattern could be preserved through several steps of reprogramming only at very limited regions. In this hypothesis, the group of 30 distally located intergenic regions could be a strong candidate. Future work is required to establish how DNA methylation is preserved in spermatozoa.

The role of noncoding RNA packaged in sperm in transgenerational inheritance could be also considered. By using a paternal mouse model in conjugation with a high-fat diet, it has recently been shown that a subset of sperm transfer RNA-derived small RNAs could generate metabolic disorders in the F1 offspring. Thus, sperm tsRNA is a paternal epigenetic factor that may mediate intergenerational inheritance (Chen et al, 2016). In another experiment, the injection of sperm RNAs from traumatized males into fertilized wild-type oocytes led to the behavioural and metabolic alterations in the offspring (Gapp et al, 2014). Thus, we can also suggest that DNA methylation alterations could affect the packaged sperm RNAs, potentially leading to interferences with other processes.

Our study revealed that gestational exposure to *thiacloprid* induces transgenerational effects on the reproductive system. We established that the *thia*-induced modifications of DNA methylation patterns in sperm are carried through generations at limited intergenic regions.

## Materials and Methods

### Ethics statement using animals

All experimental procedures were authorized by the Ministry of National Education and Research of France (Number APAFIS#17473-2018110914399411 v3). The animal facility used for the present study is licensed by the French Ministry of Agriculture (Agreement D35-238-19). All experimental procedures followed the ethical principles outlined in the Ministry of Research Guide for Care and Use of the Laboratory Animals and were approved by the local Animal Experimentation Ethics Committee (C2EA-07). All methods were in accordance with ARRIVE guidelines. Most of the animals were euthanized by placing the animals in a carbon dioxide ($CO_2$) chamber. For hormone measurement analysis, mice were euthanized

using 130 mg/kg/body weight of ketamine (Virbac) and 13 mg/kg/body weight of xylazine/Rompun (Elanco). After blood was taken from the heart, mice were euthanized via decapitation. The euthanasia procedures were done according to Annexe IV of low 2013-118 issued by the Ministry of Agriculture, Food and Forestry of France in February 1, 2013.

## Mouse treatment and dissection

Pregnant, outbred, Swiss (RjOrl), female mice were treated with *thia* (0.06, 0.6, and 6 mg/kg/day) by administering the compound down the esophagus and into the stomach using a gavage needle from embryonic day E6.5 until E15.5, which corresponds to the window of the germline population establishment. Thiacloprid (R1628-100MG; Fluka) was suspended in olive oil and administered in a volume of 150 µl for each mouse. The control mice were treated with the same volume of oil. These control and *thia*-treated mice are called F0. The progeny of exposed mice is referred to as F1. Both control and exposed F1 generation males were crossed with non-littermate, untreated females to give rise to F2 generation. Similar, both control and exposed male progenies of F2 were crossed with non-littermate and untreated females to give rise to F3 generation. For each dose, a minimum of 10 unrelated pregnant F0 female mice were treated, and 4–10 males from different litters were used for each assay. The whole experiments from F0 to F3 breeding were performed twice. Thus, animals from each generation were derived from two randomly chosen independent treatments. F1 and F3 generation males derived from treated and control groups were euthanized at day 35 for testis analysis and at day 60 for sperm and testosterone analysis. We also dissected embryonic males on the 15th day after vaginal plugs forming.

Testes and epididymis were dissected and immediately frozen in liquid nitrogen to avoid protein and RNA degradation and stored at −80°C for further analysis. One testis from a 35-d-old animal was fixed in paraformaldehyde and embedded in paraffin blocks, and some testes were used for preparation of spreads. Some epididymides were pierced, placed in Eppendorf tubes with DMEM, and incubated at 37°C for 1 h for release of mobile spermatozoa.

## Morphological analysis and immunofluorescence of paraffin sections

For morphological analysis and immunofluorescence experiments, the testes from six control and six *thia*-treated groups were fixed in 4% (wt/vol) PFA solution for 16 h, dehydrated, and embedded in paraffin. The sections were cut with a microtome with a 5-µm thickness. The sections were deparaffinized, rehydrated, and stained with hematoxylin and eosin (H&E) according to standard protocols. Images were taken with a NanoZoomer, and quantitative analysis was performed using ImageJ. Images with tubules showing the presence of all cell types, including the presence of elongated spermatids (stage VI or VII), were used for cell counts using ImageJ application. For seminiferous tubule stage classification, we used the binary decision key for staging in mice described previously (Meistrich & Hess, 2013). Cells were enumerated and divided by the total area of the tubule. We counted cells in a minimum of seven tubules using four biological replicates. For the analyses of each

testis section, stages with all cell types (spermatocytes, round spermatids, and some elongated spermatocytes) of tubules were selected, and the total area of each tubule was measured in square micrometers using ImageJ. Cells were counted in a quarter of the section of each tubule, and the result was expressed as the number of cells per 1,000 $\mu m^2$, obtained by applying the formula: n= (number of cells counted*4/total area of the tubule) *1,000. The data were averaged and plotted compared with the control group (±SD). Statistical significance was assessed with a nonparametric test.

For the lumen size, we measured the diameter in the centre of all tubules. We used round and unbroken tubules for analysis. We measured diameters in all tubules found in each section disregarding their stage. The data were averaged for each replicate and presented as lumen size compared with control. Statistical significance was assessed with a nonparametric test.

For immunofluorescence, the epitopes were unmasked in 0.01 M citrate buffer, pH 6, at 80°C for 20 min. After the sections were blocked in BSA containing 1X PBS-0.05% Tween (PBS-T), they were stained with antibodies against mouse anti-γH2AX (1:500, 05-636; Millipore) or mouse anti-PCNA (1:500, ab29; Abcam). The sections were incubated with a primary antibody overnight at 4°C in a humidified chamber. After washes in PBS-Kodak (0.04%), the sections were incubated with the appropriate fluorescent Alexa-conjugated secondary antibody (1:1,000; Invitrogen) for 1 hour in a humidified chamber at room temperature. The sections were mounted using Vectashield solution containing 0.001% (vol/vol) 4,6-diamidino-2-phenylindole dihydrochloride (DAPI). Images were taken using an AxioImager microscope equipped with an AxioCam MRc5 camera and AxioVision software version 4.8.2 (Zeiss) with a 20, 40 or 63X objective lens. We analysed cells in the *pachytene–diplotene* stage where γH2AX localized mainly in sex chromosomes, in *leptotene–zygotene* (strong staining all over the nucleus), and in round spermatids without γH2AX staining (round cells). We analysed a minimum of 10 tubules from four control and four "6" mg/kg/day *thia*-exposed animals. Cells at the corresponding stage were enumerated and presented as the average number of cells per tubule ±SD. Statistical significance was assessed with a nonparametric test.

For quantitative analysis of PCNA-positive cells, we counted cells which were positive for PCNA (*leptotene–zygotene* and *pachytene–diplotene* cells) and we compared this number with a total number of cells using DAPI-stained images.

## Spermatozoa extraction and quantification

Epididymis collected from 60-d-old mice were used for spermatozoa quantification. Briefly, epididymides from 10 control mice and 10 *thia*-derived mice were suspended in 1 ml of 0.05% Triton buffer, incised using small scissors, and homogenized for 45 s using a homogenizer (PT 2500 E; POLYTRON). The homogenate was transferred to a 15-ml conical tube and stored on ice. One milliliter aliquot of 0.05% Triton-X100 buffer was added individually to the original homogenizing vessel, homogenized, and added to the original homogenate volume. This process was repeated five times for each sample to completely obtain all spermatozoa in a final volume of 6 ml. Spermatozoa from each sample were then

enumerated using a hemocytometer. Data were obtained by taking the average of 10 squares from each biological replicate, plotted in Excel, and presented as the average number of spermatozoa per epididymis compared with the control. Seven to 10 biological replicates were used for each group, and pairwise Mann–Whitney comparisons were applied to establish statistical significance.

## Meiotic surface spread preparation for DSBs and defective chromosomal synapsing assays

Meiotic surface spreads were prepared from testes of 35-d-old mice according to a previously described protocol (Peters et al, 1997). Samples were placed in PBS, pH 7.4, at room temperature. The *tunica albuginea* was removed, and extra tubular tissue was removed by washing the seminiferous tubules with PBS. Tubules were placed in a hypotonic extraction buffer containing 30 mM Tris, 50 mM sucrose, 17 mM trisodium citrate dihydrate, 5 mM EDTA, 0.5 mM DTT, and 0.5 mM PMSF, pH 8.2, for 30–60 min. A suspension was created in 40 µl of 100 mM sucrose, pH 8.2, on a glass slide. Tubules ~2–3 cm in length were broken into pieces with forceps in 20 ml of sucrose solution. The volume was increased to 40 µl, and a 10-µl pipette was used to mix the solution until it became cloudy. The tubular remains were removed, and each suspension was divided into two clean glass slides that had just been dipped in 1% paraformaldehyde solution containing 0.15% Triton X-100. The cell suspension was then placed in a corner of the slide and dispersed across the entire slide while exposing the cells to the fixative. Slides were then washed four times for 1 min each in 0.4% Kodak Photo-Flo in PBS and dried at room temperature. The slides were then stored at −80°C until use.

Meiotic surface spread slides were incubated in a humidity chamber for 20 min at 37°C with a 1X antibody-dilution buffer (ADB) buffer containing 1% donkey serum, 0.3% BSA, and 0.005% Triton X-100 in PBS. Primary antibodies, anti-SYCP3 (1:100, sc-74569; Santa Cruz), and anti-DMC1 (1:100, sc-22768; Santa Cruz), or anti-SYCP3 (1:100, sc-74569; Santa Cruz) and anti-SYCP1 (1:100, ab15087; Abcam), or anti-SYCP3 (1:100, sc-74569; Santa Cruz) and anti-H3K9me3 (1:500, ab8898; Abcam) were then diluted in 1X ADB solution. Slides were then incubated at 4°C overnight with 50 µl of primary antibody solution with a coverslip. Two five-minute washes were then performed with 0.4% Kodak Photo-Flo in PBS. The secondary antibodies Alexa 594 goat anti-mouse and Alexa 488 chicken anti-rabbit were diluted 1:250 in 1X ADB solution. Slides were then incubated at room temperature for 20 min with 50 µl of secondary antibody solution with a coverslip. Two five-minute washes were then performed with 0.4% Kodak Photo-Flo in PBS followed by two washes with 0.4% Kodak Photo-Flo in water. Slides were air-dried, and then, 35 µl of Vectashield Mounting Medium with DAPI (Vector Laboratories) was added to each slide with a coverslip to stain nuclear DNA. Slides were then stored at −20°C until use.

DSBs were visualized with DMC1 fluorescence, which is a single-stranded DNA-binding recombinase protein involved in repairing DSBs during meiotic homologous recombination (Li et al, 1997). SYCP3 (synaptonemal complex protein 3 fluorescence) was used to visualize chromosome structure because this complex is involved in synapsis, recombination, and segregation of meiotic chromosomes (Kobayashi et al, 2017). Early *pachytene* cells have a higher

number of breaks, and we used a method described previously (Page et al, 2012) for discrimination of *pachytene* cells from *diplotene* cells based on the synapsing pattern of sex chromosomes. The synapsed sex chromosomes in the early *pachytene* stage start to desynapse in the mid *pachytene* stage. DMC1 foci were counted in all autosomes and sex chromosomes separately in a minimum of 15 cells, and the average number of foci for four biological replicates per group was compared for all tested groups. The statistical significance was assessed using the Mann–Whitney test.

Anti-rabbit SYCP1 and anti-mouse SYCP3 were also used for the examination of synaptonemal complexes to observe synapsing defects on chromosomes. A minimum of 35 images of cells in the *pachytene* stage were taken using a fluorescence microscope from a minimum of four biological replicates from the 6 mg/kg/day group and the control group. Incomplete synapsing and multiple connections (more than one chromosome joined together) were considered synapsing defects. Cells were examined and scored accordingly. Normal chromosomes were those that were fully synapsed, and SYCP1 and SYCP3 were fully colocalized, except for sex chromosomes, where only partial colocalization was detected. Cells containing more than one type of defect were scored in each defect category present. The statistical significance was assessed using a Mann–Whitney test.

Anti-rabbit H3K9me3 (1:500, ab8898; Abcam) and anti-mouse SYCP3 (1:200, sc-74569; Santa Cruz) were also used for the examination of H3K9me3 intensity over the chromosomes. The images were taken using an AxioImager microscope equipped with an AxioCam MRc5 camera and AxioVision software version 4.8.2 (Zeiss) with a 63x objective lens using the same exposure time (DAPI, 350/442, Alexa Fluor 488, 488/525, Alexa Fluor 594, 594/617). The images were left unprocessed before analysis. We quantified fluorescence inside nuclei and analysed a minimum of 30 cells in pachytene stage, from at least four different mice. To consider the background, we drew manually a region of interest in a cell-free area for each picture using imageJ. We measured the fluorescence in nucleus for DAPI and H3K9me3 and the average intensity of background region for each channel was multiplied by the area of the cell and then subtracted from the total cell fluorescence. The H3K9me3 signal was normalized for DAPI signal and the averaged values for each biological replicate were averaged and presented as normalized corrected total cell fluorescence compared with control.

## RNA extraction and RT–qPCR

Tissues stored at −80°C from the control group and from the 6 mg/kg/day treatment group were used for RNA extraction. Extraction was performed using the RNeasy Plus Mini Kit (QIAGEN). Approximately 30 mg of testis or cerebellum of adult mice were used; embryonic testes from three embryos were pooled for each biological replicate. Test samples were lysed and subsequently homogenized using a TissueLyser (QIAGEN) and 5-mm stainless steel beads (69989; QIAGEN). DNA was removed by passing the solution through a DNA elimination column. One volume of 70% ethanol was added to the lysate to provide optimal binding conditions. The lysate was then loaded onto an RNeasy silica membrane. RNA was treated with DNase using RNase-free DNase (79254; QIAGEN) directly on the column. RNA was then washed with RW1 and RPE from the RNeasy

Plus Mini Kit (QIAGEN) to remove impurities. Purified RNA was then eluted in 50 µl of RNase-free water. Six to eight biological replicates of control or treated groups were used for RT–qPCR. RT was performed using 1 µg of total RNA with iScript (1708891; Bio-Rad) adhering to the Minimum Information for Publication of Quantitative Real-Time PCR Experiments (MIQE) guidelines (Bustin et al, 2009). A no-RT reaction was performed and analysed by qPCR for all the tested genes. The reactions showed no presence of PCR products. We used *Rpl37a* as a housekeeping gene because it showed no variation between replicates based on RNA-seq data. The data are presented as the fold change compared with the control ±SD. Primers for this study were selected using the Primer-Blast program from https://www.ncbi.nlm.nih.gov/tools/primer-blast/, and most of them included exon-to-exon junctions. The primers used in this study are listed in Table S12. A nonparametric Mann–Whitney test was used for statistical significance.

### RNA sequencing and data processing

We used three biological replicates from the control group and three for each *thia* group of mice. Each replicate contained the pool of testes from three embryos because of the small size of this organ. DNA was extracted using an RNeasy kit as described above. One microgram of total RNA was used for a strand-specific library preparation protocol using NEBNext Ultra II Directional RNA Library according to the protocol provided by the manufacturer. Quality control and genome-wide sequencing were performed at the GenomEast platform at the Institute of Genetic, Molecular and Cellular Biology (IGBMC), Strasbourg, France. The sequencing was performed in massive parallel sequencing using paired-end mode, and the size of the sequencing tag was 100 bp.

Reads in FASTQ format were processed for quality control using the FastQC tool (http://www.bioinformatics.babraham.ac.uk/projects/fastqc/). An average of 50 million sequencing reads per sample was processed. The exact number of reads per sample is provided in Table S13. The reads were mapped to the reference genome [*Mus musculus* Ensembl mm10 sequence] using the Hisat2 alignment program (Kim et al, 2015) with the following parameters: reference genome file *mm10*, paired-end mode, reverse strand. The alignment files were generated as BAM files. These files were used as the input for FeatureCounts (Liao et al, 2014) to calculate the gene abundance using the gene annotation file *mm10.refGene.gtf* from UCSC. Differential gene expression was assessed using the package DESeq2 (2.11.40.7) (Love et al, 2014) with the option filtering genes with low counts. Functional annotation of DEGs was performed with the DAVID program using "Biological process" mouse gene sets using as background the list the genes that express in embryonic E15 testis.

### DNA extraction

DNA extraction was performed using a DNeasy Blood & Tissue kit (69506; QIAGEN) with modifications. First, spermatozoa were physically disrupted with a tissue lyser (QIAGEN in AL buffer) using Tungsten Carbide Beads (69997; QIAGEN). DTT (10 mM) was added to the lysis solution. Incubation with lysis solution was performed at 56°C overnight. The DNA extraction protocol

included an RNase A (19101; QIAGEN) treatment step to eliminate contaminating RNA. The DNA concentration was measured using the QuantiFluor dsDNA system (E2670; Promega). The quality of the DNA was assessed by running samples on a 0.7% agarose gel; a homogenous high-molecular weight signal was observed for each sample. DNA extraction from embryonic testes was performed according to DNeasy Blood & Tissue kits (69506; QIAGEN) without modifications.

### Chromatin immunoprecipitation (ChIP)

We performed ChIP using rabbit polyclonal antibodies against H3K9me3 (ab8898; Abcam). Equal amounts of material (~1/3 testes from one mouse) were used and incubated in 1% paraformaldehyde solution for 10 min to crosslink proteins to DNA. 100 µl of 1.25 M glycine were added to each sample to quench the unbound paraformaldehyde. The samples were centrifuged, and 1 ml of PBS and two stainless steel beads (69989; QIAGEN) were added to the pellet which was homogenized using TissueLyser (QIAGEN). Then, the samples were filtered in a cell strainer and the resulting solution was pelleted and resuspended in the following buffer: 0.25% (vol/vol) Triton X-100, 10 mM EDTA, 0.5 mM EGTA, 10 mM Tris pH8. Samples were centrifugated at 1,100*g* for 5 min, 4°C and the pellets, containing cells, were resuspended in 300 µl of SDS lysis buffer (1% [wt/vol] SDS, 10 mM EDTA, and 50 mM TrisCl pH8) in the presence of a protease inhibitor. Chromatin was sonicated in SDS lysis buffer at 60% amplitude for 8 min (20 s on, 20 s off), using Qsonica 700 sonicator (Q700-110; Newtown) supplied with cup horn 431C2; these parameters allow obtaining ~300-bp chromatin fragments. After sonication, samples were centrifugated at 14,654*g* for 10 min at 4°C and the supernatant containing sonicated chromatin was transferred and diluted in 1.7 ml of the following buffer: 0.01% (1.1% [vol/vol] Triton X-100, 1.2 mM EDTA, 16.7 mM Tris–HCl, 167 mM NaCl). A solution containing 20 µl of dynabeads (10002D; Invitrogen) and an 0.7 µl of anti-H3K9me3 antibody (ab8898; Abcam) was added to the sample tubes and incubated overnight at 4°C. Before adding the antibody and dynabeads, 10 µl of each sample was collected as "Input samples" (starting material). After overnight incubation with dynabeads and the antibody of interest, the beads were washed 5 min each, with the following four buffers: (1) low salt buffer: 0.1% (wt/vol) SDS, 1% (vol/vol) Triton X-100, 2 mM EDTA, 20 mM TrisHCl, 150 mM NaCl; (2) high salt buffer: 0.1% (wt/vol) SDS, 1% (vol/vol) Triton X-100, 2 mM EDTA, 20 mM TrisCl pH8, 500 mM NaCl; (3) LiCl buffer: 0.25 M LiCl, 1% (vol/vol) Igepal, 1 mM EDTA, 10 mM TrisCl, pH8, 1% (wt/vol) deoxycholic acid; (4) TE buffer (two washes). After the washing steps, the beads were resuspended two times in 50 µl of 1% (wt/vol) SDS, 0.1 M NaHCO3 pH9, and incubated at 65°C for 15 min to elute the precipitated chromatin from the beads. Subsequently, the eluted chromatin was reverse crosslinked by adding 9 µl of 5N NaCl and incubating at 65°C for 4 h. Then, proteins were removed by adding 1 µl 20 mg/ml of proteinase K and incubating the samples for 1 h at 45°C. The precipitated DNA was purified using MinElute Reaction Clean-Up kit (28206; QIAGEN) and the DNA concentration was measured using QuantiFluor dsDNA system (Promega). A minimum of ~5 ng of DNA was obtained.

## ChIP–qPCR

Equal amounts of precipitated DNA and input samples were used for the qPCR analysis. Quantitative PCR was performed using 0.4 ng of DNA of immunoprecipitated or Input DNA and 6 biological replicates. Normalized expression values were calculated with CFX Manager program using a region located far from promoter as a reference gene, we used a region in *Rplp0* for H3K4me3-ChIP normalization. Primers used in this study are listed in Table S14. Enrichment of each target in the precipitated DNA was evaluated by calculating the ratio between the average of the normalized ChIP DNA copies and the average of the normalized DNA copies in the inputs.

## Methylated DNA precipitation (MEDIP) and MEDIP–qPCR

For DNA methylation analysis in spermatozoa and embryonic testes, the EpiMark Methylated DNA Enrichment Kit (#E2600S; NEB) was used. A total of 6,000 ng of DNA (spermatozoa) or 3,000 ng (embryonic testis) was sonicated using a Qsonica sonicator with the following parameters: efficiency, 60%; total sonication time, 8 min, 20 s "on" and 20 s "off." The average size of sonicated DNA was 300 bp. The sonicated methylated DNA was precipitated using MBD2a-protein A-coated beads, the methylated DNA-MBD2a-protein A–coated bead complex was washed, and the DNA was eluted with elution buffer. The DNA concentration was determined according to the level of fluorescence produced by a dsDNA-binding dye (Promega). ~7–10 ng spermatozoa, ~3–4 ng (embryonic testis) of methylated DNA was recovered after precipitation, and the unprecipitated sonicated starting material was used as the input.

For MEDIP–qPCR, enriched DNA and input were diluted to equal concentrations, and qPCR was performed. Primers were designed using the coordinates of differential peaks (Table S14). The sequences were retrieved from USCSC repeat masks, and Primer-Blast was used to design primers. Equal amounts of enriched DNA and input were used for PCR. The background was normalized to the *Rplp0* unmethylated region. We used six replicates for the control and six for *thia*, and each replicate contained DNA from E15.5 testes. The data for each gene were normalized and compared with the control.

## Functional annotation of DEGs

Functional annotation of DEGs was performed by DAVID (Sherman et al, 2022) using as background the list the genes that express in embryonic E15 testis.

## A library preparation and MEDIP-seq analysis

Equal amounts of methylated DNA and input (7 ng) were taken for library preparation. Sequencing libraries were prepared using the NEBNext Ultra DNA Library Prep Kit for Illumina (E7645S; NEB). Fifteen cycles were used for library amplification. Sequencing was performed on an Illumina HiSeq4000 sequencer using a single-end 50-base read in multiplexed mode. Adapter dimer reads were removed using DimerRemover. The reads were mapped to the reference genome mm10 using Bowtie2 v2.5.0 (Langmead et al, 2009) using mm10 reference genome file, and sensitive end-to-end mode. The numbers of mapped reads were normalized by a scale factor to adjust the total number of reads. From the aligned reads, DNA methylated peaks were identified using 19 biological replicates and the corresponding input by the MACS2 (2.2.7.1) algorithm (Zhang et al, 2008); the following parameters were applied: a shift-size window of 73 bp, no model, and a q-value threshold <0.05. For comparison of the control datasets of the *thia*-exposed and control samples, differential peaks were identified using counting reads at each peak using bedtools MultiCovBed (Version 2.30.0) (Quinlan & Hall, 2010). Statistical significance was calculated using Limma v3.50.1 with filtering peaks with low counts. We performed functional annotation of the differential peaks using the web-based tool GREAT v3.0.0 (default parameters). For the visualization of ChIP-seq tracks, each bam file was converted to BedGraph tracks by using Genome Coverage 2.30.0. IGV was used to visualize the tracks. The superenhancer datasets for testis, ES cells, and embryonic day 14.5 brain were uploaded from http://www.licpathway.net/sedb/ (Wang et al, 2023) and were converted from mm9 to the reference mm10 coordinates using the LiftOver tool from UCSC. The regions were compared using bedtools intersect intervals. The exact number of sequencing reads for MEDIP-seq per sample is provided in Table S15.

## Functional annotation of DMRs

Functional annotation of DMRs was performed by GREAT using default parameters. Some figures for the illustration of differential methylated regions were generated by EaSeq (Lerdrup et al, 2016).

## The analysis of retroelements within DMRs

We downloaded the coordinates of the retroelements from UCSC browser (https://genome.ucsc.edu) and extracted the coordinates of IAPs. We used option *closest* from *bedtools* to reveal the presence of the nearest IAP element at the common 30 regions.

## Histone extraction

Histone extraction was conducted to analyse epigenetic mark occupancy using Western blotting techniques. Extraction was performed using a histone extraction kit (ab113476; Abcam) according to the protocol provided by the manufacturer. Testes stored at −80°C were used for histone extraction. Briefly, testes were homogenized with a TissueLyser (QIAGEN) and 5-mm stainless-steel beads (69989; QIAGEN) and cell extracts were pelleted via centrifugation. The pelleted cells were then suspended in a pre-lysis buffer and incubated on ice for 10 min. Cells were pelleted again with centrifugation, resuspended in three volumes of lysis buffer, and incubated on ice for 30 min. The supernatant was collected and 0, 3 volumes of DTT (ab113476) were added. Histone protein concentrations were quantified using an optical density (OD) reading at 600 nm using Pierce's solution.

## Western blot

Western blots were performed using a rabbit 1:10,000 anti-H3K9me3 polyclonal antibody (ab8898; Abcam). Equal amounts of protein extracts (10 μg) in 10 mM Tris buffer and Laemmli 4X buffer were

denatured and ran on a 4–15% gradient SDS–PAGE gel (Mini-PROTEAN TGXTM Precast Protein Gels). Proteins were transferred onto polyvinylidene difluoride membranes (Millipore) using an electro-blotter system (TE77X; Hoefer) for 1.15 h. Blocking was conducted using a 5% milk in 1X TBS Tween 0.05%. The primary antibody was diluted in 10 ml of the blocking solution and the membranes were incubated overnight at 4°C. After three ten-minute washes with 1X TBS, each membrane was incubated for 1 h in 40 ml of the blocking solution containing the corresponding HRP-conjugated secondary antibodies (1:10,000; GE Healthcare). After another three ten-minute washes with 1X TBS, Western blotting detection reagents Amersham ECL Prime Western Blotting Detection Reagent (RPN2232; Amersham) were used to coat each membrane. Specific protein expression for the antibody was then detected and measured using a molecular imager (ChemiDocTM XRS+ System with Image LabTM Software). Ponceau Red-stained bands were used to assess protein loading and normalize the levels of H3K9me3 for each sample. The intensity of the bands was measured using Fiji: ImageJ software.

### Testosterone measurement by ELISAs

Blood was taken from the hearts of deeply anesthetized animals (~800–1,000 µl). The blood was allowed to clot by leaving it undisturbed at room temperature. This process usually takes 15–30 min. We removed the clot by centrifuging for 10 min in a refrigerated centrifuge. The resulting supernatant was preserved at −80°C until use. We used a mouse testosterone ELISA kit (orb340113-960; CliniSciences) for testosterone measurement. We followed the instructions provided by the manufacturer. The data were averaged, plotted, and presented as the concentration of testosterone in ng/ml.

### Statistical analyses

We used the minimum number of animals according to the requirements of the EU Ethics Committee. The number of animals used was specified for each experimental procedure. We performed the Kruskal–Wallis test to assess statistical significance in body weight and spermatozoa number measurements, a nonparametric Wilcoxon–Mann–Whitney for qPCR experiments, immunofluorescence, synapsing defects, DMC1 foci, testosterone quantifications, and WB assay.

## Data Availability

All raw and processed sequencing data generated in this study have been submitted to the NCBI Gene Expression Omnibus (GEO; https://www.ncbi.nlm.nih.gov/geo/) under accession numbers GSE234861 and GSE235213.

## Supplementary Information

## Acknowledgements

Sequencing was performed by the GenomEast platform, a member of the "France Génomique" consortium (ANR-10-INBS-0009). We would also like to thank the Galaxy platform for bioinformatics support (Galaxy Community, 2022). We would also like to thank the H2P2 platform (UMS Biosit Inserm UMS 018, CNRS UMS3480) for assistance with the preparation and analysis of paraffin testis sections. We are grateful to Christian Jaulin for critical reading and suggestions for the article. This work was supported by ANSES funding (R20155NN) to F Smagulova.

### Author Contributions

O Dali: methodology and writing—review and editing.
S D'Cruz: methodology.
L Legoff: methodology.
M Diba Lahmidi: methodology and writing—review and editing.
C Heitz: methodology and writing—review and editing.
P-E Merret: methodology and writing—review and editing.
P-Y Kernanec: methodology.
F Pakdel: investigation and writing—review and editing.
F Smagulova: data curation, formal analysis, investigation, methodology, and writing—original draft, review, and editing.

### Conflict of Interest Statement

The authors declare that they have no conflict of interest.

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
