## [Reviewer comments · Life Science Alliance]

Life Science Alliance

Transgenerational epigenetic effects imposed by neonicotinoid thiacloprid exposure

Fatima Smagulova, Ouzna Dali, Shereen D'Cruz, Louis Legoff, Mariam Diba Lahmidi, Celine Heitz, Pierre-Etienne Merret, Pierre-Yves Kernanec, and Farzad Pakdel

DOI: <https://doi.org/10.26508/lsa.202302237>

Corresponding author(s): Fatima Smagulova, Délégation Régionale Grand Ouest

Review Timeline:

Submission Date:	2023-06-26
Editorial Decision:	2023-08-07
Revision Received:	2023-10-02
Editorial Decision:	2023-10-23
Revision Received:	2023-11-06
Accepted:	2023-11-07

Transaction Report:

August 7, 2023

Re: Life Science Alliance manuscript #LSA-2023-02237-T

Dr. Fatima Smagulova
Inserm/IrsetU1085
9 avenue du Prof. Léon Bernard
Rennes 35000

Dear Dr. Smagulova,

Thank you for submitting your manuscript entitled "Ancestral gestational exposure to the widely used neonicotinoid thiacloprid leads to global DNA methylation alterations in spermatozoa in three generations of male mice.". The manuscript has been evaluated by expert reviewers, whose reports are appended below. Unfortunately, after an assessment of the reviewer feedback, our editorial decision is against publication in Life Science Alliance.

Although your manuscript is intriguing, I feel that the points raised by the reviewers are more substantial than can be addressed in a typical revision period. If you wish to expedite publication of the current data, it may be best to pursue publication at another journal.

Given the interest in the topic, I would be open to re-submission to Life Science Alliance of a significantly revised and extended manuscript that fully addresses the reviewers' concerns and is subject to further peer review. If you would like to resubmit this work to Life Science Alliance, you may submit an appeal directly through our manuscript submission system. Please note that priority and novelty would be reassessed at re-submission.

Regardless of how you choose to proceed, we hope that the comments below will prove constructive as your work progresses.

Thank you for thinking of Life Science Alliance as an appropriate place to publish your work.

Sincerely,

Reviewer #1 (Comments to the Authors (Required)):

In this manuscript, Dali et al characterize the transgenerational effects of prenatal exposure to the neonicotinoid thiacloprid (thia) by examining spermatogenesis phenotypes, gene expression, and DNA methylation effects in the F3 generation following exposure. This study extends a previous paper by the same group characterizing the direct effects of prenatal thia exposure in the F1 generation (Hartman et al., 2021). Phenotype in F3s is described based on sperm count, testis weight, relative fraction of spermatogenic cell populations, as well as markers of DNA recombination and repair and of heterochromatin in adult testis and meiotic spreads. Gene expression and DNA methylation studies are done in embryonic testes at E15.5 and in sperm (for DNA methylation). The experiment and dataset are interesting and have the potential to be highly valuable, combining phenotype, expression, and DNA methylation data from a transgenerational model of chemical exposure. However, as presented there are significant weaknesses with the study and manuscript that prevent exciting conclusions from being drawn. First and most fundamentally, the breeding design is not well explained (further detailed below) making it impossible to evaluate the likely sources of noise in the data. Second, the genome-wide data is under-analyzed; as shown, the quality and significance of these datasets are difficult to evaluate. In some cases the rigor and robustness of individual experiments could be improved by better specifying the experimental procedures and by adding data. All of these weaknesses together make it difficult to understand the major conclusions or model suggested by the data and therefore to determine if the claims are supported. Thoroughly discussing the study design to exclude fundamental flaws, improving the experimental support provided for the claims and performing a more rigorous analysis of the existing dataset would substantially improve this potentially interesting study. The paper should also be better framed so that its significance is clearer. Finally, there are several basic mistakes in statements about germ cell development and epigenetic regulation which must be corrected.

General concerns:

1) Study design: it is not clear exactly how F3 males are obtained from prenatally exposed F1s. Are F1 and F2 offspring bred

among themselves, or outbred to naïve mates? Where do the control animals come from in this scheme? The details of the breeding scheme are especially important for two reasons. First, the study was done in outbred (Swiss) mice. Therefore, there might be substantial genetic effects between litters of interbred animals that could explain the effect being attributed to thia exposure. It is important to know where controls come from, especially RNA-seq and DNAm experiments. Second, if females used to generate F2s and F3s have been previously exposed to thia, this is a major confounding factor and the observed effects cannot be unambiguously assigned to inheritance through the male germ line. It would also be helpful to summarize in Figure S1 which tissues and time points are taken at which generation.

2) Related to point 1 above, the effects reported are generally small, highly variable, or both. To some extent this is expected in transgenerational epigenetic inheritance. However it is difficult to interpret the biological significance of small and variable effects without having more precise knowledge of the potential sources of genetic and epigenetic variability. For qPCR experiments especially (1E, 3C, 6B, 6C), there is a large amount of variability between data points. Biological but not technical replicates are mentioned in the methods. Given the variability, if technical replicates were also done (multiple wells on the same plate from the same sample), this should be explicitly specified to increase confidence in the data.

3) In several places, discussion of the relationship of different epigenetic marks to each other and to gene expression is either excessively vague or inaccurate. Examples are the discussion of DNA methylation and super-enhancers (page 5 lines 3-13), relationship between H3K9me3 and retroelement expression (page 11 line 6-7), relationship between gene expression and DNA methylation (page 18 line 20-29), and the relationship between DNA methylation and H3K4me3 (page 21 line 2-5).

4) The Introduction does not clearly explain the rationale and significance of the study. The previous Hartman et al. study should be introduced earlier to provide context, and the gap in knowledge more clearly laid out.

Evaluation of specific claims:

- Support for spermatogenic defects in F3 mice: data are moderately supportive. Additional analyses and experiments could be completed within 1-2 months.

5) In Fig 1D, higher magnification examples of the gH2AX staining used to identify each stage should be provided; as shown, it is difficult to assess the accuracy of the method.

6) In Figure 1E, if RT-qPCR was performed on whole testis (which is not specified in the legend or main text), the changes in cell composition reported in Fig. 1D should be reflected in expression of cell type markers detected by qPCR. To separate the effects on gene expression from effects on spermatogenic progression, it would be better to do the qPCR experiment in sorted cells or juvenile testes in the first wave. An additional issue is that the effects on expression by qPCR appear to be in the opposite direction expected based on the effects on cell composition, which is difficult to explain.

7) There should be some explanation provided for how the spermatogenic defects observed are expected to relate to the 29% reduced sperm count. Is reduced sperm count expected to be a downstream effect of the differences in meiotic progression, or a separate defect?

- Support for altered heterochromatin and retroelement expression: data are not strongly supportive. Additional analyses and experiments could be completed within 1-2 months.

8) In Fig 3A, the methodology for quantitation of H3K9me3 immunofluorescence should be explained in detail. In general, quantification of immunofluorescence levels in meiotic spreads is unreliable. Overall changes in levels of H3K9me3 could be more robustly supported by immunofluorescence in tissue sections and/or Western blotting in whole testes.

9) H3K9me3 normally acts to repress retroelements, so it is surprising that F3 testes apparently have an increase in H3K9me3 but also an increase in retroelement expression. In the Hartman 2021 study, F1s show decreased H3K9me3 and increased retroelement expression, as might be expected. A suggested explanation for this effect should be provided.

10) All the data is already available in this study to evaluate a relationship between changes in DNA methylation and changes in retroelement expression. This is an established relationship (retroelements are repressed by DNA methylation), so the authors should perform an analysis to determine if retroelements are affected by changes in DNA methylation in this system.

- Support for gene expression changes in F3 mice: data are not strongly supportive. Additional analysis of this data could be completed within 2 months.

11) For Gene Ontology enrichments in Fig. 4, the background used to calculate enrichments should be specified. If the whole genome is used as background, finding enrichment of germ cell functions just reflects the tissue being used. It would be better to use the set of all genes expressed in E15.5 testes.

12) In general, it is difficult to fully evaluate or interpret the RNA-seq data because the analysis is superficial; nothing is shown from this data except the GO enrichments. Intermediate analyses such as volcano plots of the DEGs should be shown to give

an idea of the magnitude of the reported effect and distribution of the data.

13) In Table 1 and Fig. 5 (DNA methylation analysis), the switch in DMR proportions from majority down (F1, F2) to majority up (F3) is surprising. Is there a reasonable model for this switch, or is it more likely to reflect noise in the dataset?

- Support for altered epigenetic state in F3s: data are not strongly supportive. Additional analyses of these data could be completed within two months.

14) Similar to the RNA-seq data, the MeDIP-seq data is superficially analyzed and displayed. This is a potentially very interesting dataset but it is very hard to draw conclusions about it as shown. The IGV tracks are extremely difficult to see at the size provided. The scale of the tracks (both base pairs and signal strength) should be shown. Volcano plots, MA plots, or correlation dot plots between samples could be used to indicate the magnitude and distribution of differential methylation. It is also hard to evaluate the robustness of the data without more analysis of data quality, especially since the qPCR validation provided (Fig. 6) is extremely variable.

Other comments:

15) This statement is not correct (page 2 line 25-26): "somatic cells undergo reprogramming and become germ cells in a process known as a somatic-to-germline transition". In mammalian systems, germ lineage is considered to remain segregated throughout the life cycle (Weissmann barrier). There is no reference provided. This should be removed or better clarified.

16) Cell division of secondary spermatocytes is not a mitotic division (Page 3, line 4-5). It is considered a second meiotic division, although it resembles mitosis in that sister chromatids segregate rather than homologous chromosomes.

17) Page 4 line 15-16: more seminal and more recent studies could be used to support the role of DNA methylation in transgenerational epigenetic inheritance, including PMID: 10545949, 12601169, and 36754048.

18) Page 4 line 18-19: Because the specific time points make a difference for the expected effects in this study, it would be useful to specify the time that PGCs reach the genital ridge (E10.5) and colonize the gonad (E11.5) and lose methyl marks (E12.5-E.13.5).

19) Page 6 line 6: It would be useful to provide a reference point for the significance of the chosen doses should be given based on exposures in humans; this information is provided in the Discussion (page 19) but would be useful to know earlier.

20) Fig. 2: The significance of numbers of DMC1 foci should be better explained in the text.

21) Fig 4: Some GO terms are shown at non-significant p-values (e.g. 5.98e-01); these should be removed.

Referee Cross-Comments:

Reviewer 2 and I are well aligned in our concerns. Furthermore, I agree with reviewer 2 that there are serious concerns about Figure 3C. On review of Figure 9F in Hartman et al., 2021, the distribution of data points is highly similar. I agree that source data should be provided for this figure and for the 2021 figure as a comparison.

Reviewer #2 (Comments to the Authors (Required)):

This study by Dali et al investigates the effect of exposure to the neonicotinoid thiacloprid (thia) in utero on DNA methylation profiles in embryonic testis and postnatal spermatozoa in subsequent generations of male mice. Previously, this research group demonstrated that thia exposure in utero caused spermatogenic defects in the F1 generation (Hartman et al, 2021) and conducted this current study as a follow-up to evaluate if similar effects could be observed in later generations. The authors examined among other outcomes gene expression and DNA methylation in embryonic testis; markers of meiosis function, H3K9me3 level and gene expression in 35-day old progeny and spermatozoa and testosterone levels in adult mice.

The main new insight is the transgenerational effect of thia exposure in altering reproductive health and epigenetic marks in F3 progeny. Since only a substantially high dose led to notable changes, there is likely little implication for human health, but the study does provide insight as a proof of concept and examines potential mechanisms and is therefore of interest to the field.

Overall, the study is conducted thoroughly and provides a wealth of information. However, this reviewer has serious concerns regards Figure 3C. The observed gene expression changes in retroelements in generation F3 look exactly like the ones reported in the previous paper by Hartman et al. Given the exact same distribution of individual data points in the treatment group, this reviewer has concerns that the same data was replotted, despite claiming to originate from a different set of experiments. The source data needs to be provided to allow for verification and correction of any possible mistakes.

(comparing current study Fig 3C from F3, with Hartman et al Fig 9; from F1 generation- doi: 10.3389/fcell.2021.691060)

The evidence presented in Figure 3 regards increase in H3K9me3 is not sufficient for the conclusions drawn, and a Western Blot should be provided. Furthermore, could the authors please discuss why retroelement expression increases if the H3K9me3 levels also increase? Given the function in transcriptional silencing (-such as the silencing of retroelements), a global increase in H3K9me3 would furthermore silence more genes. Thus, it is expected to represent changes in distribution of H3K9me3, e.g. reduced marking in retroelements with increase at other loci. If feasible, chromatin-immunoprecipitation data could enrich the authors' findings here.

The conclusions as stated in abstract and in the main text are not sufficiently justified. If the hypothesis was true, that the transgenerational effect is mediated via preserved alterations in DNA methylation (i.e. resistant to germ cell reprogramming), the observed effects would have to be the same or similar in all 3 generations, and the effects could be observed in any F3 tissue. Whilst around 500 genes had DNA methylation changes to some degree in sperm of all 3 generations, this was not necessarily in the same direction. Hence, the phenotypic outcomes in F3 are likely not the result of preserved epigenetic marks but could be caused by other factors derived from the initial pathology in F1, which in turn affects gene regulation in following generations.

Minor comments:

All figure legends need to specify n-numbers, meaning of error bars and statistical tests used.

To improve clarity, all figure legends should indicate which exact tissue and at which age was sampled so that it is not necessary to look for the required information in the text.

Figure resolution needs to be increased e.g. Fig 5A-B, as the shown example IGV tracks are difficult to see and the effect size hard to judge.

This reviewer is surprised by high number of differentially expressed genes in F3 testes. The provided gene ontology provides little insight given the lack of specificity. Therefore, some data on quality control and inter-sample variation should be provided to test for artifacts such as caused by one sample outlier.

Please provide further rationale for analyzing super-enhancers of unrelated tissues, and some statistical indication if this represents an enrichment over random regions.

We are submitting the amended version of our manuscript. We are very thankful for a giving us an opportunity to resubmit our manuscript to *Life Science Alliance* journal. We also express our sincere gratitude to the anonymous Reviewers for their careful reading, critics and comments. We feel that their suggestions and comments improved our manuscript! We addressed most of their comments and points. In the amended version, we changed the introduction as it was suggested, we removed or modified wrong statements, and we added missing references.

We added several experiments as it was suggested by Reviewers.

- 1) We added another germ cell marker analysis, PCNA, this marker was preferentially detected in meiotic cells. We performed a quantitative analysis of this marker (Figure 1E-F).
- 2) We performed H3K9me3 western blot analysis from purified histones as it was suggested (Figure 3C-D). WB confirmed our previously observed increase in H3K9me3 by immunofluorescence.
- 3) We performed ChIP using H3K9me3 antibody and analysed the occupancy of this histone at regions known to be enriched in this mark (Figure 3E).
- 4) We replaced wrong Figure 3F RT-qPCR of retroelements in F3 with correct one.
- 5) We performed a bioinformatic analysis using RNA-seq data. Specifically, we added volcano plots in main figure (Figure 4A and Figure 4C) to show the distribution of differentially expressed genes. We added principal component analysis (PCA), dispersion heatmaps and MA-plots to a Supplementary figure S5. These graphs show high variation between biological replicates in F1 and less in F3 embryos. We performed functional annotation of DEGs using as background the list of the genes that express in embryonic E15 testis (Figure 4B and Figure 4D). We confirmed several targets identified by RNA-seq in embryonic E15 testis in F3 by RT-qPCR (Figure 4E).
- 6) We confirmed DNA methylation changes at targets identified by MEDIP-seq by performing MEDIP-qPCR analysis.
- 7) We changed Figure 5A by replacing plots with higher resolution and we added the sequencing read scales on Figure 5A. We added PCA to illustrate the variation between samples and volcano plots to show the distribution of differentially methylated regions (supplementary Figure S8).
- 8) We also identified a group of 36 genes which have similar directions in changes in DNA methylation (Supplementary table S2).
- 9) We calculated the estimates of overrepresentation of brain SE in sperms

In the discussion we added some sentences about possible mechanisms that change spermatozoa numbers in F3. We discuss how persistence of double strand breaks could affect meiosis. We also proposed some mechanisms that could lead to decrease in spermatozoa number in F3 males.

We hope that our manuscript is now suitable for publication in *Life Science Alliance* journal.

Sincerely,

Fatima Smagulova

Bellow we addressed point-by-point the comments

First and most fundamentally, the breeding design is not well explained (further detailed below) making it impossible to evaluate the likely sources of noise in the data.

We agree that it is very important to carefully describe the breeding strategy. In a new version, we explained the details of the breeding strategy in "Mouse treatment and dissection" in Methods section.

Thiacloprid (Fluka, R1628-100MG) was suspended in olive oil and administered in a volume of 150 μ L for each mouse at dose “0”, “0.6” or “6”. The control mice were treated with the same volume of oil. These control and thia-treated mice are called F0. Progeny of exposed mice is called F1. Both control and exposed F1 generation males were crossed with non-littermate, untreated female to give rise to F2 generation. Both control and exposed male progeny of F2 were crossed with non-littermate and untreated females to give rise to F3 generation. For each dose, a minimum of 10 unrelated pregnant F0 female mice were treated, and for each assay 4-10 males from different litters were used. The whole experiments from F0 to F3 breeding we performed twice. Thus, animals from each generation were derived from two randomly chosen independent treatments. F1 and F3 generation males derived from treated and control groups were euthanized at day 35 for a testis analysis and at day 60 for a spermatozoa analysis. We also dissected embryonic males at 15th day after vaginal plugs forming. Similar F3 embryos were sacrificed. Thus, control mice were treated by oil and *thia* mice with thiacloprid their progeny were crossed until third generation.

1) Study design: it is not clear exactly how F3 males are obtained from prenatally exposed F1s. Are F1 and F2 offspring bred among themselves, or outbred to naïve mates? Where do the control animals come from in this scheme? The details of the breeding scheme are especially important for two reasons. First, the study was done in outbred (Swiss) mice. Therefore, there might be substantial genetic effects between litters of interbred animals that could explain the effect being attributed to thia exposure.

We explained the details of breeding in the previous comment. Yes, we agree that outbred mice are genetically heterogenous and thereby the use of them lead to increased biological variation. Outbred mouse strain is used in many toxicological studies because outbred mice better model in toxicological studies to a very heterogenous human population. Since goal of our toxicological study is to reveal the effects that could human have, we opted to use outbred mice. However, the disadvantage of outbred mice use is a large biological variation between replicates.

It is important to know where controls come from, especially RNA-seq and DNase experiments. Second, if females used to generate F2s and F3s have been previously exposed to thia, this is a major confounding factor and the observed effects cannot be unambiguously assigned to inheritance through the male germ line. It would also be helpful to summarize in Figure S1 which tissues and time points are taken at which generation.

We explained the details of breeding in a previous comment. All females except F0 were not related not littermate and to clearly follow the paternal inheritance, for each new generation cross, we purchase the age-matched females. In Figure S1 we specified which tissue was used and for which experiment.

2) Related to point 1 above, the effects reported are generally small, highly variable, or both. To some extent this is expected in transgenerational epigenetic inheritance. However it is difficult to interpret the biological significance of small and variable effects without having more precise knowledge of the potential sources of genetic and epigenetic variability. For qPCR experiments especially (1E, 3C, 6B, 6C), there is a large amount of variability between data points. Biological but not technical replicates are mentioned in the methods. Given the variability, if technical replicates were also done (multiple wells on the same plate from the same sample), this should be explicitly specified to increase confidence in the data.

For all our experiments we did not use technical but biological replicates. All our mice progeny were created from minimum 10 independent mothers which were treated either with oil either with oil plus pesticide. Large variations in our work are due to fact that we use outbred mouse strain. This strain is

used in many toxicological studies as it better fits to very heterogenous human population. Since goal of our toxicological study is to reveal the effects that humans could have, we opted to use outbred mice. However, the disadvantage of outbred mice use is large biological variations between replicates.

We followed the advice of the Referee N1 and we repeated experiment with RT-QPCR in testis of 35-day-old animals using technical replicates (three replicate for each gene). Our new RT-QPCR confirmed the previous result, all the previously determined significantly changed genes were significant in a new RT-QPCR analysis, we added this data in a new updated figure 1G.

3) In several places, discussion of the relationship of different epigenetic marks to each other and to gene expression is either excessively vague or inaccurate. Examples are the discussion of DNA methylation and super-enhancers (page 5 lines 3-13), relationship between H3K9me3 and retroelement expression (page 11 line 6-7), relationship between gene expression and DNA methylation (page 18 line 20-29), and the relationship between DNA methylation and H3K4me3 (page 21 line 2-5).

We removed or corrected wrong statements. We are very thankful for Reviewer for these comments.

4) The Introduction does not clearly explain the rationale and significance of the study. The previous Hartman et al. study should be introduced earlier to provide context, and the gap in knowledge more clearly laid out.

The previous study was presented earlier and observed effects were presented as it was suggested. We also added a statement what is major goal of our study.

Evaluation of specific claims:

- Support for spermatogenic defects in F3 mice: data are moderately supportive. Additional analyses and experiments could be completed within 1-2 months.

We added now the analysis of PCNA marker. We observed a decrease in meiotic germ cells in seminiferous tubules.

5) In Fig 1D, higher magnification examples of the gH2AX staining used to identify each stage should be provided; as shown, it is difficult to assess the accuracy of the method.

In the new version the higher magnification and explanations are provided.

6) In Figure 1E, if RT-qPCR was performed on whole testis (which is not specified in the legend or main text), the changes in cell composition reported in Fig. 1D should be reflected in expression of cell type markers detected by qPCR. To separate the effects on gene expression from effects on spermatogenic progression, it would be better to do the qPCR experiment in sorted cells or juvenile testes in the first wave.

For the analysis of the testis defects, we used mice which were 35 days old. We chose this age that is corresponding to the end of first wave of spermatogenesis.

An additional issue is that the effects on expression by qPCR appear to be in the opposite direction expected based on the effects on cell composition, which is difficult to explain.

We observed similar increase in *Rad51* expression in both F1 and F3. In F3, we observed several genes that specific for spermatids were upregulated in F3 males (*Prm1*, *Spaca1*, and *Tsk1*).

7) There should be some explanation provided for how the spermatogenic defects observed are expected to relate to the 29% reduced sperm count. Is reduced sperm count expected to be a downstream effect of the differences in meiotic progression, or a separate defect?

We believe that the synapsing meiotic defects and persistence of DSBs could lead to a reduction in gamete production. As it was previously shown, p53 and TAp63 participate in the recombination-dependent pachytene arrest and apoptosis in mouse spermatocytes (Marcet-Ortega et al. 2017). The persistence of DMC1 foci were observed in mice model knockout experiments with ablation of gene encoding meiotic specific proteins such as Meiob (Souquet et al. 2013) and Tex19.1 (Crichton et al. 2018). In these cases meiosis was arrested at the zygotene/pachytene stage of prophase I and cells failed to progress further and were eliminated. We can similarly suggest that there is meiotic failure in *thia* progeny of mice that could lead to meiotic arrest and apoptosis that explain the decrease in spermatozoa.

- Support for altered heterochromatin and retroelement expression: data are not strongly supportive. Additional analyses and experiments could be completed within 1-2 months.

This point raised by secondary Referee too, thus, we did several experiments. We performed WB analysis of H3K9me3 using testis tissue of 35-day-old males and we observed there is increase in H3K9me3 in F3 which is consistent with our immunofluorescence (IF) data. We also performed ChIP using H3K9me3 antibody and we analysed H3K9me3 level at several targets which are normally enriched in this mark: imprinted genes, retroelements, satellite DNA and telomeres repeats. We observed the increase in H3K9me3 which is consistent with IF and WB data.

8) In Fig. 3A, the methodology for quantitation of H3K9me3 immunofluorescence should be explained in detail. In general, quantification of immunofluorescence levels in meiotic spreads is unreliable. Overall changes in levels of H3K9me3 could be more robustly supported by immunofluorescence in tissue sections and/or Western blotting in whole testes.

We added the details of quantitation of H3K9me3 immunofluorescence to Methods section. The images were taken using an AxioImager microscope equipped with an AxioCam MRc5 camera and AxioVision software version 4.8.2 (Zeiss, Le Pecq, France) with a 63X objective lens using the same exposure time (DAPI, 350/442, Alexa Fluor 488, 488/525, Alexa Fluor 594, 594/617). The images were left unprocessed prior to analysis. We quantified fluorescence inside nuclei and analyzed a minimum of 30 cells in pachytene stage, from at least 4 different mice. To consider the background, we drew manually a region of interest in a cell-free area for each picture. We measured the fluorescence in nucleus for DAPI and H3K9me3 and the averaged intensity of background region for each channel was multiplied by the area of the cell and then subtracted from the total cell fluorescence. The H3K9me3 signal was normalized for DAPI signal and the averaged value for each biological replicate were averaged and presented as normalized corrected total cell fluorescence compared to control.

9) H3K9me3 normally acts to repress retroelements, so it is surprising that F3 testes apparently have an increase in H3K9me3 but also an increase in retroelement expression. In the Hartman 2021 study, F1s show decreased H3K9me3 and increased retroelement expression, as might be expected. A suggested explanation for this effect should be provided.

The provided figure was an error from our part, we copy-pasted a wrong figure. Now, we provide a correct data. Indeed, we observed a slight but not significant decrease in retroelement expression. However, at IAP show tendency to increase in expression. DNA methylation is not the only mechanisms that control retroelement expression. Small RNAs such as piwi RNAs also play role in the control of retroelements gene expression level. It is tempting to think that a reduced activity of piwi RNAs could lead to compensatory higher activity of lysine methyltransferases that could contribute to the increased level of H3K9me3.

10) All the data is already available in this study to evaluate a relationship between changes in DNA methylation and changes in retroelement expression. This is an established relationship (retroelements are repressed by DNA methylation), so the authors should perform an analysis to determine if retroelements are affected by changes in DNA methylation in this system.

We would like to note that retroelements RNA expression analysis was done in the 35-day-old testes but MEDIP was done in spermatozoa, between testis and spermatozoa there are several remodelling processes, such as histone -to-protamine transition and some other alterations occur. Nevertheless, we performed MEDIP -qPCR analysis in sperm for retroelements and we observed that in sperm of F3 there is increase in DNA methylation but in testis of F3 there is not significant change in retroelements expression.

- Support for gene expression changes in F3 mice: data are not strongly supportive. Additional analysis of this data could be completed within 2 months.

We chose to confirm several genes detected in F3 RNA-seq by RT-qPCR. We extracted RNA from e15.5 and performed RT-qPCR analysis. At global level, the changes in gene expression are similar between RNAseq and RT-qPCR. However, two methods use a different way of normalization. Personally, I believe that RNA-seq is more precise as it takes into account large number of sequencings reads for quantification, while RT-QPCR is dependent on where primers are located, if primers are not located at the place of major form of expressed transcripts, RT-qPCR may not be accurate as RNA-seq method.

Besides that, we provide volcano plots and heatmap and MA-plots for F1 and F3 RNA-seq. In our plots we can see we have higher dispersion in F1 than in F3 in biological replicates that can explain the fewer DEGs were detected in F1.

11) For Gene Ontology enrichments in Fig. 4, the background used to calculate enrichments should be specified. If the whole genome is used as background, finding enrichment of germ cell functions just reflects the tissue being used. It would be better to use the set of all genes expressed in E15.5 testes.

We are very grateful for this suggestion! We followed the advice and performed functional annotation of DEGs using as a background list the genes that are expressed in E15 testis.

12) In general, it is difficult to fully evaluate or interpret the RNA-seq data because the analysis is superficial; nothing is shown from this data except the GO enrichments. Intermediate analyses such as

volcano plots of the DEGs should be shown to give an idea of the magnitude of the reported effect and distribution of the data.

We added Volcano plots, MA-plots and PCA plots to Supplementary Figure S5. We observed a higher variation between replicates in F1 compared to F3. In our previous transgenerational studies, a similar higher variation in F1 compared to F3 was also observed. We cannot simply explain this phenomenon. We observed more DEGs in F3.

13) In Table 1 and Fig. 5 (DNA methylation analysis), the switch in DMR proportions from majority down (F1, F2) to majority up (F3) is surprising. Is there a reasonable model for this switch, or is it more likely to reflect noise in the dataset?

Indeed, we observed the DMRs switch to opposite in F3. F1 and F2 are directly exposed mice, F1 generation mice were exposed *in utero* to toxicant. The germ cells of F1 generation mice were also exposed to toxicant. These cells will be precursors for future spermatozooids for F2. Thus, F3 is a first generation which is not directly expose to thiacloprid. We can imagine that some compensatory mechanisms would play role in F3 generation. In spite that that 90 % of shared peaks showed a different direction in the change, there is still 10% of common peaks which corresponds to 36 genes which have similar changes in directions. We are thinking that those regions contain genes that could be a master regulator of other 90%, because these regions are located near important developmental genes such as *Sox2* and transcriptional factors *Foxn1*, *Foxd1* and *Isl1*.

- Support for altered epigenetic state in F3s: data are not strongly supportive. Additional analyses of these data could be completed within two months.

We performed new MEDIP followed by qPCR and we found that most of qPCR data are consistent with observed MEDIP-seq changes.

14) Similar to the RNA-seq data, the MeDIP-seq data is superficially analyzed and displayed. This is a potentially very interesting dataset but it is very hard to draw conclusions about it as shown. The IGV tracks are extremely difficult to see at the size provided. The scale of the tracks (both base pairs and signal strength) should be shown. Volcano plots, MA plots, or correlation dot plots between samples could be used to indicate the magnitude and distribution of differential methylation. It is also hard to evaluate the robustness of the data without more analysis of data quality, especially since the qPCR validation provided (Fig. 6) is extremely variable.

We followed the advice of Referee and we added volcano plots and PCA analysis of MEDIP-seq data to supplementary figure S8. We also changed the size of our figure and we indicated the scale of the sequencing signals in IGV figures. We also repeated MEDIP-seq in embryonic F1 testis. The embryonic testes are very small; to perform this experiment we have to pool testes from several embryos in order to get one replicate. This probably increases biological variation. We very carefully repeated the experiment using embryonic material and now we provided new data which have less variation.

Other comments:

15) This statement is not correct (page 2 line 25-26): "somatic cells undergo reprogramming and become germ cells in a process known as a somatic-to-germline transition". In mammalian systems, germ lineage is considered to remain segregated throughout the life cycle (Weissmann barrier). There is no reference provided. This should be removed or better clarified.

We removed the wrong statement

16) Cell division of secondary spermatocytes is not a mitotic division (Page 3, line 4-5). It is considered a second meiotic division, although it resembles mitosis in that sister chromatids segregate rather than homologous chromosomes.

We corrected that phrase for ... “Then, each secondary spermatocyte undergoes second meiotic division to give rise to haploid spermatids”.

17) Page 4 line 15-16: more seminal and more recent studies could be used to support the role of DNA methylation in transgenerational epigenetic inheritance, including PMID: 10545949, 12601169, and 36754048.

We added references and changed the introduction.

18) Page 4 line 18-19: Because the specific time points make a difference for the expected effects in this study, it would be useful to specify the time that PGCs reach the genital ridge (E10.5) and colonize the gonad (E11.5) and lose methyl marks (E12.5-E.13.5).

Many thanks for this suggestion! We added the references and changed statements

19) Page 6 line 6: It would be useful to provide a reference point for the significance of the chosen doses should be given based on exposures in humans; this information is provided in the Discussion (page 19) but would be useful to know earlier.

We added the significance of the chosen doses in Experimental design Results part.

20) Fig. 2: The significance of numbers of DMC1 foci should be better explained in the text.

We believe that the synapsing meiotic defects and persistence of DSBs could lead to a reduction in gamete production. As it was previously shown, p53 and TAp63 participate in the recombination-dependent pachytene arrest and apoptosis in mouse spermatocytes (Marcet-Ortega et al. 2017). The persistence of DMC1 foci were observed in mice model knockout experiments with ablation of genes encoding meiotic specific proteins such as Meiob (Souquet et al. 2013) and Tex19.1 (Crichton et al. 2018). In these cases meiosis was arrested at the zygotene/pachytene stage of prophase I and cells failed to progress further and were eliminated. We can similarly suggest that there is meiotic failure in *thia* progeny of mice that could lead to meiotic arrest and apoptosis that explain the decrease in spermatozoa.

21) Fig 4: Some GO terms are shown at non-significant p-values (e.g. 5.98e-01); these should be removed.

We have now replaced this figure with new one

Referee Cross-Comments:

Reviewer 2 and I are well aligned in our concerns. Furthermore, I agree with reviewer 2 that there are serious concerns about Figure 3C. On review of Figure 9F in Hartman et al., 2021, the distribution of data points is highly similar. I agree that source data should be provided for this figure and for the 2021 figure as a comparison.

We apologise for our mistake, now we added correct figure and we also cited a previous F1 results.

Reviewer #2 (Comments to the Authors (Required)):

This study by Dali et al investigates the effect of exposure to the neonicotinoid thiacloprid (thia) in utero on DNA methylation profiles in embryonic testis and postnatal spermatozoa in subsequent generations of male mice. Previously, this research group demonstrated that thia exposure in utero caused spermatogenic defects in the F1 generation (Hartman et al, 2021) and conducted this current study as a follow-up to evaluate if similar effects could be observed in later generations. The authors examined among other outcomes gene expression and DNA methylation in embryonic testis; markers of meiosis function, H3K9me3 level and gene expression in 35-day old progeny and spermatozoa and testosterone levels in adult mice.

The main new insight is the transgenerational effect of thia exposure in altering reproductive health and epigenetic marks in F3 progeny. Since only a substantially high dose led to notable changes, there is likely little implication for human health, but the study does provide insight as a proof of concept and examines potential mechanisms and is therefore of interest to the field.

Overall, the study is conducted thoroughly and provides a wealth of information. However, this reviewer has serious concerns regards Figure 3C. The observed gene expression changes in retroelements in generation F3 look exactly like the ones reported in the previous paper by Hartman et al. Given the exact same distribution of individual data points in the treatment group, this reviewer has concerns that the same data was replotted, despite claiming to originate from a different set of experiments. The source data needs to be provided to allow for verification and correction of any possible mistakes.

(comparing current study Fig 3C from F3, with Hartman et al Fig 9; from F1 generation- doi: 10.3389/fcell.2021.691060)

We deeply apologize that by mistake we copy-past a wrong graph. We are very thankful to the Referee to note that. We now provide the correct graph in a Figure 3F. In F3 testis the retroelement expression did not change.

The evidence presented in Figure 3 regards increase in H3K9me3 is not sufficient for the conclusions drawn, and a Western Blot should be provided. Furthermore, could the authors please discuss why retroelement expression increases if the H3K9me3 levels also increase? Given the function in transcriptional silencing (-such as the silencing of retroelements), a global increase in H3K9me3 would furthermore silence more genes. Thus, it is expected to represent changes in distribution of H3K9me3, e.g. reduced marking in retroelements with increase at other loci. If feasible, chromatin-immunoprecipitation data could enrich the authors' findings here.

Many thanks for this suggestion! We performed ChIP using H3K9me3 and we analysed H3K9me3 in several regions known to be enriched in this mark; imprinted genes, retroelements, satellite DNA and telomeric repeats. We observed the increase in H3K9me3 in all tested regions, our new data is consistent with IF and WB data.

The conclusions as stated in abstract and in the main text are not sufficiently justified. If the hypothesis was true, that the transgenerational effect is mediated via preserved alterations in DNA methylation (i.e. resistant to germ cell reprogramming), the observed effects would have to be the same or similar in all 3 generations, and the effects could be observed in any F3 tissue. Whilst around 500 genes had DNA methylation changes to some degree in sperm of all 3 generations, this was not necessarily in the same direction. Hence, the phenotypic outcomes in F3 are likely not the result of preserved epigenetic marks but could be caused by other factors derived from the initial pathology in F1, which in turn affects gene regulation in following generations.

We took a closer look at the regions with altered DNA methylation in F1 and F3. Despite the fact that the vast majority did not have the same direction in changes, we identified ~30 regions with similar directions in changes in F1 and F3. GREAT assigned these regions to 36 genes that have similar changes in DNA methylation, mainly decrease. These include many developmental genes (*Isl1*, *Lama1*, *Tle1*, *Sox2*, *Foxn1*, *Foxd4*, *Inha*, *Zfp64*). Thus, it is possible that, that the vast majority of DMRs regions might be regulated by other mechanisms but there are still some regions which could resist reprogramming and preserved DNA methylation.

Minor comments:

All figure legends need to specify n-numbers, meaning of error bars and statistical tests used.

We added now the number of animals, statistical tests and meaning of error bars to the figure legends.

To improve clarity, all figure legends should indicate which exact tissue and at which age was sampled so that it is not necessary to look for the required information in the text.

We added the origin of tissue, the age of animals and statistical tests to the figure legends.

Figure resolution needs to be increased e.g. Fig 5A-B, as the shown example IGV tracks are difficult to see and the effect size hard to judge.

We added higher magnification as it was also suggested by Referee number 1

This reviewer is surprised by high number of differentially expressed genes in F3 testes. The provided gene ontology provides little insight given the lack of specificity. Therefore, some data on quality control and inter-sample variation should be provided to test for artifacts such as caused by one sample outlier.

We added new graphs such as Volcano plots and PCA analysis, Data in F3 looks less dispersed and less variation was observed

Please provide further rationale for analysing super-enhancers of unrelated tissues, and some statistical indication if this represents an enrichment over random regions.

We used nonrelevant somatic tissue to see whether information of other tissue was impacted or only germ cell. We chose to analyse SE active in germ cells (testis) and somatic cells (brain) and precursors

of these both tissues (ES). We found that statistically overrepresentation of brain SE is less $p < 10e-6$, hypergeometric test.

October 23, 2023

RE: Life Science Alliance Manuscript #LSA-2023-02237-TR-A

Dr. Fatima Smagulova
Délégation Régionale Grand Ouest
Inserm/Irset U1085
9 avenue du Prof. Léon Bernard
Rennes 35000
France

Dear Dr. Smagulova,

Thank you for submitting your revised manuscript entitled "Transgenerational effects of neonicotinoid thiacloprid on DNA methylation in murine spermatozoa.". We would be happy to publish your paper in Life Science Alliance pending final revisions necessary to meet our formatting guidelines.

- please address the Reviewers' remaining comments
- please upload all figure files as individual ones, including the supplementary figure files; all figure legends should only appear in the main manuscript file
- please add the Twitter handle of your host institute/organization as well as your own or/and one of the authors in our system
- abstract should be a single paragraph not exceeding 175 words
- please incorporate any points from the Conclusion section into the Discussion; we only allow a Discussion section
- please move your main, supplementary figure, and table legends in the main manuscript text after the references section
- we encourage you to revise the figure legends for figures one and S2 such that the figure panels are introduced in alphabetical order;
- please add callouts for Figures S2C-F; S4A-B; S5A-B; S7A-L; S8A-C and Table S14 to your main manuscript text
- please upload your Tables in editable .doc or Excel format
- please indicate the scale bar size in Legend for Figure S3

Figure Checks:

- please add scale bars to the microscopy images in Figures 1, 2 and 3, and indicate their sizes in the corresponding figure legend

A. FINAL FILES:

-- Summary blurb (enter in submission system): A short text summarizing in a single sentence the study (max. 200 characters including spaces). This text is used in conjunction with the titles of papers, hence should be informative and complementary to the title. It should describe the context and significance of the findings for a general readership; it should be written in the

present tense and refer to the work in the third person. Author names should not be mentioned.

B. MANUSCRIPT ORGANIZATION AND FORMATTING:

Sincerely,

Reviewer #1 (Comments to the Authors (Required)):

The authors have been very responsive to reviews and this manuscript is significantly improved. Most importantly, the breeding scheme has been clarified and the experimental setup is confirmed to be appropriate, and the problematic duplicate figure panel (previously Figure 3C) has been replaced. In addition, analysis and presentation of the RNA-seq and meDIP-seq datasets is substantially expanded and improved, Western blots have been added to support statements about H3K9me3 levels, additional replicates for qPCR assays have been added to improve robustness of the results, and some incorrect statements in the introduction and results sections have been corrected and clarified. I have a few minor concerns remaining that should be straightforward to address:

- Line 46: Specify mouse testis: "in the embryonic testis" -> "in the mouse embryonic testis"
- The discussion of super-enhancers in lines 60-68 seems disjointed as it stands. The updated discussion of SEs in the relevant part of the Results section is much improved and could stand alone. I suggest removing the paragraph from the introduction and limiting the explanation of SEs to the results section.
- The fact that transmission is being assayed by breeding only through the male line after thia exposure, crossing to wild type unexposed females, is a critical aspect of the study and should be briefly specified in the first paragraph of the results section.
- There remains a minor concern about the use of 35d whole testes for Fig. 1G (previously 1E), since changes in germ cell marker expression levels assayed by RT-qPCR could reflect cell composition instead of cell-intrinsic transcriptional changes. This concern is minor because the direction of change detected for most markers is different from what would be predicted if the changes were due to cell composition, based on the changes in germ cell numbers reported in Fig. 1C-F. However, I think this possible artifact should be briefly acknowledged in the text.
- Line 185-186: "These data suggest a possible impact of thia exposure on both meiotic and postmeiotic populations of cells".

This could be made slightly more specific: "These data suggest a possible impact of thia exposure *on gene expression* in both meiotic and postmeiotic populations of cells"?

- In Fig. 3D, loading control normalization is not mentioned in the legend or in the methods. Was band intensity normalized to the Ponceau stain shown in Supp. Fig. S4? This should be specified.

- In Fig 4B and D: it would be helpful to label on the figure which GO enrichment plot is from upregulated and which from downregulated genes.

- In Fig. S8C (MEDIP-seq volcano plot for F3s), something seems funny. There is an accumulation of values at a specific p-value, which suggests that there might be an artifact in the data processing. If not an artifact, can a brief explanation be provided in the text or legend?

- Line 326-327: Details about bedtools could be moved to the Methods section.

- Line 433: There's a possible typo: these -> thia

- In general, please increase the font size for the axis labels across most of the plots, especially the ones that were newly added in this revision, since in some cases it is very hard to read the axis scales and labels.

Reviewer cross-comments:

I fully agree with the comments of the other reviewer.

Reviewer #2 (Comments to the Authors (Required)):

Overall, the authors have made sufficient efforts to address the main concerns raised by the reviewer. Specifically, the main concern regards data duplication was addressed and the mistake was corrected. Whilst showing a very different result, those corrected results are more plausible, since they are in line with the expected silencing function of H3K9me3. In addition, the authors provide the requested Western Blot and the suggested ChIP.

Furthermore, data on RNA-seq quality control has been provided. It confirms this reviewers' concerns that inter-sample variation of biological replicates, particularly in F1, substantially exceeds the effect size of the treatment (Fig S5). Therefore, the RNA-seq data is difficult to interpret. However, the authors acknowledge this aspect.

There has been an improvement in the labelling and visibility of the figures.

In their rebuttal letter, the authors respond to the discussion points mentioned, but it is not obvious where and how they modified the manuscript accordingly. This information would have been helpful. Additionally, the manuscript would benefit from proof-reading as well as revision of language and organization (such as confining experimental details to the method section, pp. 9,11)

Dear Editors!

Thank you very much for a consideration of our manuscript. Please find our responses to the comments of Referees and Editors!

The Referee's Comments

Reviewer #1 (Comments to the Authors (Required)):

The authors have been very responsive to reviews and this manuscript is significantly improved. Most importantly, the breeding scheme has been clarified and the experimental setup is confirmed to be appropriate, and the problematic duplicate figure panel (previously Figure 3C) has been replaced. In addition, analysis and presentation of the RNA-seq and meDIP-seq datasets is substantially expanded and improved, Western blots have been added to support statements about H3K9me3 levels, additional replicates for qPCR assays have been added to improve robustness of the results, and some incorrect statements in the introduction and results sections have been corrected and clarified. I have a few minor concerns remaining that should be straightforward to address:

We are very grateful for the appreciation of our work! It is encouraging!

- Line 46: Specify mouse testis: "in the embryonic testis" -> "in the mouse embryonic testis"

We modified that, p2 line 44

- The discussion of super-enhancers in lines 60-68 seems disjointed as it stands. The updated discussion of SEs in the relevant part of the Results section is much improved and could stand alone. I suggest removing the paragraph from the introduction and limiting the explanation of SEs to the results section.

We modified accordingly, p12 line 323

- The fact that transmission is being assayed by breeding only through the male line after thia exposure, crossing to wild type unexposed females, is a critical aspect of the study and should be briefly specified in the first paragraph of the results section.

We specified that "breeding was done only through the male line after thia exposure", p108 line 109

- There remains a minor concern about the use of 35d whole testes for Fig. 1G (previously 1E), since changes in germ cell marker expression levels assayed by RT-qPCR could reflect cell composition instead of cell-intrinsic transcriptional changes. This concern is minor because the direction of change detected for most markers is different from what would be predicted if the changes were due to cell

composition, based on the changes in germ cell numbers reported in Fig. 1C-F. However, I think this possible artifact should be briefly acknowledged in the text.

We specified that the analysis was done in whole testis, p7, line 168

- Line 185-186: "These data suggest a possible impact of thia exposure on both meiotic and postmeiotic populations of cells". This could be made slightly more specific: "These data suggest a possible impact of thia exposure *on gene expression* in both meiotic and postmeiotic populations of cells"?

We modified the sentence to "These data suggest a possible impact of thia exposure *on gene expression* in both meiotic and postmeiotic populations of cells", p7, line 173.

- In Fig. 3D, loading control normalization is not mentioned in the legend or in the methods. Was band intensity normalized to the Ponceau stain shown in Supp. Fig. S4? This should be specified.

We specified that "The band intensity was normalized to the Ponceau stain shown in Fig S4B", p8, line 110

- In Fig 4B and D: it would be helpful to label on the figure which GO enrichment plot is from upregulated and which from downregulated genes.

We indicated on the figures which plots are from upregulated and which from downregulated genes (Fig 4B and Fig4D).

- In Fig. S8C (MEDIP-seq volcano plot for F3s), something seems funny. There is an accumulation of values at a specific p-value, which suggests that there might be an artifact in the data processing. If not an artifact, can a brief explanation be provided in the text or legend?

This was an artefact of copy-paste. We now fixed that (Fig. S8D-F)

- Line 326-327: Details about bedtools could be moved to the Methods section.

We moved the text to the Methods section, p28, line 746.

- Line 433: There's a possible typo: these -> thia

Yes, this was a typo, we corrected that, p16, line 416.

- In general, please increase the font size for the axis labels across most of the plots, especially the ones that were newly added in this revision, since in some cases it is very hard to read the axis scales and labels.

We increased the font size of the labels.

Reviewer cross-comments:

I fully agree with the comments of the other reviewer.

Reviewer #2 (Comments to the Authors (Required)):

Overall, the authors have made sufficient efforts to address the main concerns raised by the reviewer. Specifically, the main concern regards data duplication was addressed and the mistake was corrected. Whilst showing a very different result, those corrected results are more plausible, since they are in line with the expected silencing function of H3K9me3. In addition, the authors provide the requested Western Blot and the suggested ChIP.

Furthermore, data on RNA-seq quality control has been provided. It confirms this reviewers' concerns that inter-sample variation of biological replicates, particularly in F1, substantially exceeds the effect size of the treatment (Fig S5). Therefore, the RNA-seq data is difficult to interpret. However, the authors acknowledge this aspect.

There has been an improvement in the labelling and visibility of the figures.

In their rebuttal letter, the authors respond to the discussion points mentioned, but it is not obvious where and how they modified the manuscript accordingly. This information would have been helpful. Additionally, the manuscript would benefit from proof-reading as well as revision of language and organization (such as confining experimental details to the method section, pp. 9,11)

Now, we highlighted the modifications. We also checked the language and corrected the mistakes.

The Editors Comments

long with points mentioned below, please tend to the following:

-please address the Reviewers' remaining comments

That is done.

-please upload all figure files as individual ones, including the supplementary figure files; all figure legends should only appear in the main manuscript file

That is done

-please add the Twitter handle of your host institute/organization as well as your own or/and one of the authors in our system

That is done, our twitter is https://twitter.com/irset_fr

-abstract should be a single paragraph not exceeding 175 words

That is done.

-please incorporate any points from the Conclusion section into the Discussion; we only allow a Discussion section

That is done.

-please move your main, supplementary figure, and table legends in the main manuscript text after the references section

That is done.

-we encourage you to revise the figure legends for figures one and S2 such that the figure panels are introduced in alphabetical order;

That is done.

-please add callouts for Figures S2C-F; S4A-B; S5A-B; S7A-L; S8A-C and Table S14 to your main manuscript text

That is done.

-please upload your Tables in editable .doc or Excel format

That is done.

-please indicate the scale bar size in Legend for Figure S3

That is done.

Figure Checks:

-please add scale bars to the microscopy images in Figures 1, 2 and 3, and indicate their sizes in the corresponding figure legend

That is done.

That is done.

Summary blurb

We exposed pregnant female mice to neonicotinoid thiacloprid. We observed reproductive system defects together with DNA methylation and histone H3K9me3 changes in the third-generation males.

November 7, 2023

RE: Life Science Alliance Manuscript #LSA-2023-02237-TRR

Dr. Fatima Smagulova
Délégation Régionale Grand Ouest
Inserm/Irset U1085
9 avenue du Prof. Léon Bernard
Rennes 35000
France

Dear Dr. Smagulova,

Thank you for submitting your Research Article entitled "Transgenerational epigenetic effects imposed by neonicotinoid thiacloprid exposure". It is a pleasure to let you know that your manuscript is now accepted for publication in Life Science Alliance. Congratulations on this interesting work.

DISTRIBUTION OF MATERIALS:

Again, congratulations on a very nice paper. I hope you found the review process to be constructive and are pleased with how the manuscript was handled editorially. We look forward to future exciting submissions from your lab.

Sincerely,
